# Cellular entry and uncoating of naked and quasi-enveloped human hepatoviruses

Efraín E Rivera-Serrano[1,2], Olga González-López[1,2], Anshuman Das[2], Stanley M Lemon[2,3]*

[1]Lineberger Comprehensive Cancer Center, The University of North Carolina at Chapel Hill, Chapel Hill, United States; [2]Department of Medicine, The University of North Carolina at Chapel Hill, Chapel Hill, United States; [3]Department of Microbiology and Immunology, The University of North Carolina at Chapel Hill, Chapel Hill, United States

**Abstract** Many 'non-enveloped' viruses, including hepatitis A virus (HAV), are released non-lytically from infected cells as infectious, quasi-enveloped virions cloaked in host membranes. Quasi-enveloped HAV (eHAV) mediates stealthy cell-to-cell spread within the liver, whereas stable naked virions shed in feces are optimized for environmental transmission. eHAV lacks virus-encoded surface proteins, and how it enters cells is unknown. We show both virion types enter by clathrin- and dynamin-dependent endocytosis, facilitated by integrin $\beta_1$, and traffic through early and late endosomes. Uncoating of naked virions occurs in late endosomes, whereas eHAV undergoes ALIX-dependent trafficking to lysosomes where the quasi-envelope is enzymatically degraded and uncoating ensues coincident with breaching of endolysosomal membranes. Neither virion requires PLA2G16, a phospholipase essential for entry of other picornaviruses. Thus naked and quasi-enveloped virions enter via similar endocytic pathways, but uncoat in different compartments and release their genomes to the cytosol in a manner mechanistically distinct from other *Picornaviridae*.

DOI: https://doi.org/10.7554/eLife.43983.001

*For correspondence:
smlemon@med.unc.edu

Competing interests: The authors declare that no competing interests exist.

## Introduction

The presence or absence of an external lipid envelope has featured strongly in the systematic classification of animal viruses for decades. However, many viruses that have previously been considered to be 'non-enveloped' are now known to be released non-lytically from infected cells in a 'quasi-enveloped' form, enclosed in small extracellular vesicles (EVs) devoid of virus-encoded surface proteins. This phenomenon was recognized first among members of the *Picornaviridae*, including hepatitis A virus (HAV, genus Hepatovirus) (*Feng et al., 2013*), poliovirus and coxsackievirus B (genus Enterovirus) (*Bird et al., 2014*; *Chen et al., 2015*; *Jackson et al., 2005*; *Robinson et al., 2014*), but it has been demonstrated also for hepatitis E virus (*Hepeviridae*), rotaviruses (*Reoviridae*), and noroviruses (*Caliciviridae*) (*Nagashima et al., 2017*; *Santiana et al., 2018*). The size of the virus-containing EVs varies widely among different viruses, as does the number of virus capsids enclosed in each vesicle, most likely reflecting different mechanisms of biogenesis. However, these membrane-wrapped, quasi-enveloped virions share the capacity to infect cells, and contribute to pathogenesis either by cloaking capsids in membranes such that they are sequestered from the host immune system, or possibly by increasing the number of viral genomes delivered to newly infected cells, thereby facilitating genetic complementation (*Chen et al., 2015*; *Feng et al., 2013*).

HAV provides a prime example of viral quasi-envelopment. An ancient pathogen that remains a common cause of enterically-transmitted hepatitis globally (*Lemon et al., 2017*), it is hepatotropic in vivo and released without cell lysis in small EVs containing 1–3 capsids (*Feng et al., 2013*). In

**eLife digest** The Hepatitis A virus is a common cause of liver disease in humans. It is unable to multiply on its own so it needs to enter the cells of its host and hijack them to make new virus particles.

Infected human cells produce two different types of Hepatitis A particles. The first, known as 'naked' virus particles, consist of molecules of ribonucleic acid (or RNA for short) that are surrounded by a protein shell. Naked virus particles are shed in the feces of infected individuals and are very stable, allowing the virus to spread in the environment to find new hosts.

At the same time, a second type of particle, known as the 'quasi-enveloped' virus, circulates in the blood of the infected individual. In a quasi-enveloped particle, the RNA and protein shell are completely enclosed within a membrane that is released from the host cell. This membrane protects the protein shell from human immune responses, enabling quasi-enveloped virus particles to spread in a stealthy fashion within the liver.

It was not clear how these two different types of virus particle are both able to enter cells despite their surface being so different. To address this question, Rivera-Serrano et al. used a microscopy approach to observe Hepatitis A particles infecting human liver cells.

The experiments showed that both types of virus particle actually use similar routes. First, the external membrane of the cell folded around the particles, creating a vesicle that trapped the viruses and brought them within the cell. Inside these vesicles, the naked virus particles soon fell apart, and their RNA was released directly into the interior of the cell.

However, the vesicles that carried quasi-enveloped virus travelled further into the cell and eventually delivered their contents to a specialized compartment, the lysosome, where the virus membrane was degraded. This caused the quasi-enveloped viruses to fall apart and release their RNA into the cell more slowly than the naked particles.

Several viruses, such as the one that causes polio, also have quasi-enveloped forms. Studying how these particles are able to infect human cells while hiding behind membranes borrowed from the host may help us target these viruses better.

DOI: https://doi.org/10.7554/eLife.43983.002

contrast to the naked, nonenveloped virus that is shed in feces, these quasi-enveloped virions (eHAV) are the only form of virus found in sera from infected humans and account for most viruses in supernatant fluids of permissive cell cultures (*Feng et al., 2013*). They are fully infectious,~50–110 nm in diameter, and possess a buoyant density of ~1.100 g/cm$^3$ in iodixanol. The protein composition of the quasi-envelope resembles that of exosomes, suggesting a multivesicular body (MVB) origin, and their biogenesis is dependent on ALIX (ALG-2-interacting protein 1, also known as PDCD6IP) and other components of the <u>e</u>ndosomal <u>s</u>orting <u>c</u>omplexes <u>r</u>equired for <u>t</u>ransport (ESCRT) (*Feng et al., 2013*; *González-López et al., 2018*; *McKnight et al., 2017*). The capsids enclosed within the eHAV vesicle are, like other picornaviral capsids, comprised of 60 copies of each of 4 proteins (*Wang et al., 2015*). However, they differ from the naked, nonenveloped capsids shed in feces in that they contain an unprocessed form of the VP1 capsid protein (VP1pX) retaining a 71 amino acid carboxy-terminal domain absent in naked virions (*Feng et al., 2013*).

Interactions between picornaviral capsids and their receptors are critical for promoting endocytosis, virion uncoating, and safe delivery of the viral RNA genome across endosomal membranes into the cytoplasm to establish a productive infection (*Baggen et al., 2018*; *Groppelli et al., 2017*; *Strauss et al., 2015*). The phosphotidylserine (PtdSer) receptor TIM1 (T cell immunoglobulin and mucin-containing domain one protein, also known as HAVCR1) was identified as a receptor for HAV twenty years ago (*Feigelstock et al., 1998*; *Kaplan et al., 1996*), prior to the discovery of quasi-enveloped virions. TIM1 has since been shown to facilitate the binding of eHAV but not naked HAV virions to the cell surface (*Das et al., 2017*), presumably through binding PtdSer displayed on the surface of the eHAV membrane (*Feng et al., 2015*). However, TIM1 is not essential for attachment or entry of either HAV or eHAV, nor is it required for infection of permissive strains of mice (*Das et al., 2017*). Thus far, an essential receptor has yet to be identified for either HAV or eHAV.

Little is known about how these two virion types enter cells, although prior studies point to the existence of distinct entry pathways for naked versus quasi-enveloped virions as might be expected from the presence of the limiting lipid membrane in eHAV. eHAV is selectively sensitive to the lysosomal poison chloroquine (*Feng et al., 2013*), and slower to enter cells and begin replication than naked HAV capsids (*Das et al., 2017*; *Feng et al., 2013*). eHAV is resistant to anti-capsid neutralizing antibodies in quantal, plaque reduction-like assays, but neutralizing antibodies restrict its replication when added to cells 4–6 hr after adsorption of the virus, suggesting a delay in uncoating within an endocytic compartment (*Feng et al., 2013*). Here, we report detailed roadmaps for the entry of these different types of infectious hepatitis A virions in hepatocytes, identify a key role for integrin $\beta_1$ in endocytosis of both, and demonstrate distinct trafficking of these virion types through the endocytic system. We demonstrate critical temporal and spatial differences in the uncoating of HAV and eHAV capsids, and show that eHAV entry is uniquely dependent on the ESCRT accessory protein ALIX as well as lysosomal proteins involved in lipid metabolism.

## Results

### Determinants of HAV and eHAV endocytosis

To identify the endocytic pathways responsible for internalization of HAV and eHAV virions in Huh-7.5 human hepatoma cells, we used pharmacological and genetic approaches to disrupt the function of regulators of several canonical endocytic routes. Inhibition of clathrin- and dynamin-mediated endocytosis by the drugs chlorpromazine and dynasore, respectively, strongly inhibited the uptake of both gradient-purified HAV and eHAV virions, as measured by RT-PCR quantitation of the viral genome in cell lysates 6 hr post-infection (hpi) (*Figure 1A*, *Figure 1—figure supplement 1*). In contrast, inhibiting cavaeolae-dependent endocytosis with filipin resulted in mild impairment of eHAV entry only, whereas inhibiting cytosolic dynein, actin- and/or Rac1-dependent micropinocytosis, or heparan sulfate proteoglycan binding with various compounds had no effect on the entry of either virion type (*Figure 1A*). Roles for clathrin and dynamin in the uptake of both HAV and eHAV were confirmed by siRNA-mediated depletion of clathrin heavy chain (*CLTC*), the µ1 subunit of the clathrin-associated adaptor complex 2 (*AP2M1*), or dynamin-2 (*DNM2*) (*Figure 1B*, *Figure 1—figure supplement 2A*). Caveolin-1 (*CAV1*) depletion minimally inhibited entry of eHAV only, consistent with the effect of filipin treatment, while depleting the clathrin- and caveolae-independent endocytosis regulators ADP ribosylation factor 6 (*ARF6*) and flotillin-1 (*FLOT1*) had no significant effect. Consistent with these results, confocal fluorescence microscopy revealed a high-degree of co-localization of both HAV and eHAV capsid antigen with clathrin-coated vesicles between ~0.5–1 hpi, and minimal co-localization of the eHAV capsid only with caveolin-1 (*Figure 1C*). Thus, both HAV and eHAV entry occur primarily through clathrin- and dynamin-dependent endocytosis, although caveolin-dependent endocytosis may play a minor role in eHAV uptake.

The host proteins associated with the eHAV quasi-envelope are similar to those identified in exosomes (*McKnight et al., 2017*). This suggests that eHAV entry might be mediated by integrins or adhesion molecules previously reported to be involved in the uptake of EVs (*van Dongen et al., 2016*). Consistent with this hypothesis, siRNA-mediated depletion of integrin $\beta_1$ (*ITGB1*) resulted in a striking and highly significant reduction in the uptake of quasi-enveloped as well as naked virions (*Figure 1D*, *Figure 1—figure supplement 2B*). Depleting integrin $\beta_1$ did not reduce the quantity of eHAV or HAV bound to the cell surface at 4°C, but significantly reduced the amount of eHAV and HAV RNA present in Huh-7.5 cells 6 hpi at 37°C (*Figure 1E*). CRISPR/Cas9 knockout of *ITGB1* also reduced both eHAV and HAV uptake and spread in H1-HeLa cells (*Figure 1F,G*). Consistent with these results, pre-treating Huh-7.5 cells with an RGD peptide containing an integrin $\beta_1$-binding motif reduced uptake of both virion types by about 50% (*Figure 1H*). On the other hand, pre-treating cells with antibodies that activate integrin $\beta_1$ by binding to and stabilizing specific $\beta_1$ conformations (*Su et al., 2016*) increased viral uptake compared to an inert integrin $\beta_1$ antibody (K-20), and revealed differences in the interaction of integrin $\beta_1$ with eHAV versus HAV (*Figure 1H*). The activating antibody TS2/16, which binds an open conformation of $\beta_1$ (*Su et al., 2016*), enhanced eHAV but not HAV entry, whereas 8E3 and HUTS-4, which bind extended and open headpiece $\beta_1$ conformations, respectively, had the opposite effect, enhancing naked HAV but not quasi-enveloped eHAV

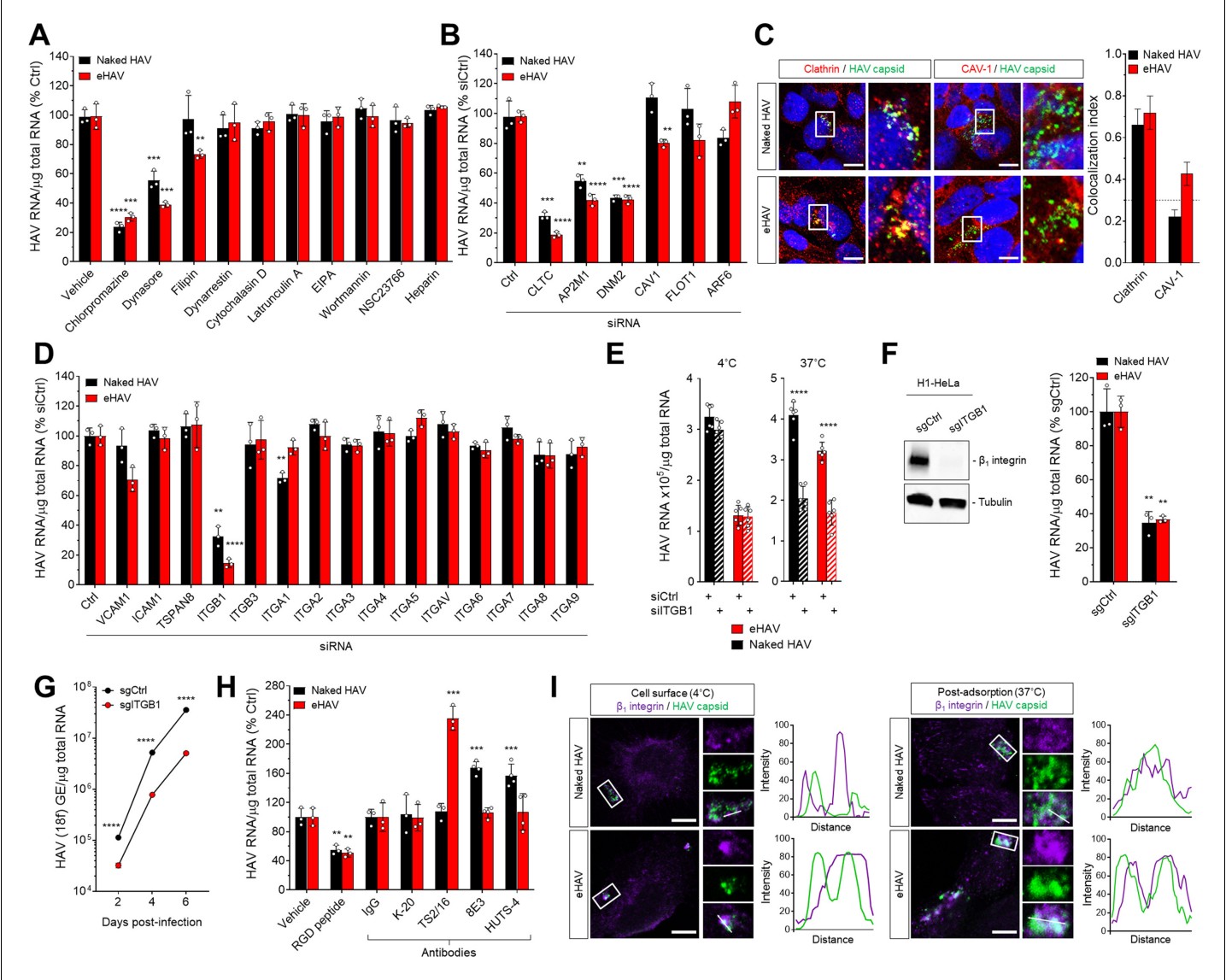

**Figure 1.** Naked and quasi-enveloped HAV virions undergo clathrin-dependent endocytosis facilitated by β1 integrin. (**A**) Effect of endocytic inhibitors on HAV and eHAV entry quantified by RT-PCR (mean ± SD, n = 3 independent experiments). (**B**) Effect of siRNA-mediated depletion of endocytic regulators on HAV and eHAV entry (mean ± SD, n = 3 independent experiments). See *Figure 1—figure supplement 2A* for knockdown efficiencies. (**C**) Confocal micrographs of Huh-7.5 cells immunolabeled with anti-HAV capsid (K24F2) and anti-clathrin or anti-caveolin-1 at one hpi. Scale bar, 10 μm. (**D**) Effect of siRNA-mediated depletion of integrins and adhesion molecules on HAV and eHAV entry (mean ± SD, n = 3 independent experiments). See *Figure 1—figure supplement 2B* for knockdown efficiencies. (**E**) Effect of β1 integrin knockdown on HAV/eHAV attachment to cells 2 hpi at 4°C (mean ± SD, n = 6 biological replicates from two independent experiments) or on HAV and eHAV uptake 6 hpi at 37°C (mean ± SD, n = 3 independent experiments). (**F**) Levels of HAV RNA at 20 hpi in H1-HeLa engineered by CRISPR/Cas9 using a control or a specific *ITGB1*-targeting sgRNA (mean ± SD, n = 3 biological replicates of a representative experiment). (**G**) Replication of rapid-replicating HAV (18 f) in sgITGB1 H1-HeLa cells (mean ± SD, n = 3). (**H**) Effect of an inhibitory RGD-containing peptide or β1 integrin activating antibodies on HAV and eHAV uptake (mean ± SD, n = 3–4 biological replicates of a representative experiment). (**I**) Confocal micrographs of Huh-7.5 cells incubated at 4°C or 37°C for 1 hr and immunolabeled with anti-HAV capsid (K24F2) and anti-β1 integrin. Scale bar, 10 μm. For numeric data plotted in graphs associated with this figure, see *Figure 1—source data 1*.
DOI: https://doi.org/10.7554/eLife.43983.003

The following source data and figure supplements are available for figure 1:

**Source data 1.** Source data corresponding to *Figure 1*.
DOI: https://doi.org/10.7554/eLife.43983.008

**Figure supplement 1.** Huh-7.5 cells were incubated with DMEM supplemented with the indicated inhibitors for 1 hr.
DOI: https://doi.org/10.7554/eLife.43983.004

**Figure supplement 2.** Immunoblots showing siRNA-mediated knockdown efficiencies corresponding to data of *Figure 1*.

*Figure 1 continued on next page*

*Figure 1 continued*

DOI: https://doi.org/10.7554/eLife.43983.005

**Figure supplement 3.** Levels of HAV RNA at 24 hpi in H1-HeLa cells engineered by CRISPR/Cas9 to be knocked-out for the indicated integrins inoculated with naked HAV (mean ± SD).

DOI: https://doi.org/10.7554/eLife.43983.006

**Figure supplement 4.** Huh-7.5 cells were inoculated with HAV or eHAV and subjected to immunostaining one hpi using antibodies against HAV capsid (K24F2) and $\alpha_5$ or $\alpha_V$ integrins.

DOI: https://doi.org/10.7554/eLife.43983.007

entry. These data hint at differences in the ligands, yet to be identified, that are bound by integrin $\beta_1$ during eHAV and HAV entry.

In contrast to the impact of integrin $\beta_1$ depletion, depletion experiments failed to confirm a requirement for any specific $\alpha$ integrin in the uptake of either virion (*Figure 1D*, *Figure 1—figure supplement 2B*). While RNAi-mediated depletion of integrin $\alpha$1 caused a modest but statistically significant decrease in HAV uptake in Huh-7.5 cells, this was not confirmed in H1-HeLa cells with CRISPR/Cas9 knockout of *ITGA1* (*Figure 1D*, *Figure 1—figure supplement 3*). Confocal microscopic imaging also suggested eHAV was associated with integrin $\beta_1$, both at the surface of Huh-7.5 cells at 4°C and during virion internalization at 37°C (*Figure 1I*), but not with either $\alpha_5$ or $\alpha_V$ integrins (*Figure 1E*, *Figure 1—figure supplement 4*). Collectively, these results demonstrate that HAV and eHAV are dependent on distinct integrin $\beta_1$ interactions for uptake by clathrin- and dynamin-mediated endocytosis, but leave unanswered the role of $\alpha$ integrins.

## Distinct intracellular trafficking routes for naked and quasi-enveloped HAV

Several GTPases are well-known for their role in the sorting of cargo through functionally distinct endosomes, with Rab5A and Rab7a involved in trafficking through early and late endosomes, respectively (*Mellman, 1996*; *Mercer et al., 2010*). Confocal microscopy of infected Huh-7.5 cells revealed transient co-localization of the capsid antigen in both naked and quasi-enveloped virions with Rab5A$^+$ and Rab7a$^+$ compartments around~1–2 hpi (*Figure 2A*). In contrast, neither type of virion was associated with Rab11A$^+$ recycling endosomes. RNAi-mediated depletion of Rab5A or Rab7a, but not Rab11A, resulted in a significant reduction in the accumulation of intracellular HAV RNA (*Figure 2B*, *Figure 2—figure supplement 1*). Thus, both types of HAV virions traffic through early and late endosomes shortly after uptake into the cell through clathrin-mediated endocytosis.

Our earlier studies suggested that infection with eHAV, but not naked HAV, requires endosomal acidification since it was specifically inhibited by the lysosomal poison, chloroquine (*Feng et al., 2013*). Consistent with this, confocal microscopy demonstrated that the capsid antigen associated with quasi-enveloped eHAV, but not naked HAV, was selectively trafficked to LAMP1$^+$ and VAMP8$^+$ lysosomes as early as 4 hpi, remaining there up to 12 hpi (*Figure 2C*, *Figure 2—figure supplement 2*). Notably, naked virion capsid antigen was never found to be associated with lysosomes, suggesting that Rab7a$^+$ late endosomes represent the final trafficking destination of naked HAV. Importantly, sorting of both eHAV and naked HAV virions was associated with co-internalization of integrin $\beta_1$ to these endolysosomal compartments (*Figure 2D*, *Figure 2—figure supplement 3*). Similar results were obtained when integrin $\beta_1$ expressed on the cell surface was labeled prior to virus adsorption with the activating monoclonal antibody TS2/16 (*Figure 2D*, *Figure 2—figure supplement 4*), which triggers the endocytosis and trafficking of integrin $\beta_1$ to lysosomes (*Margadant et al., 2012*). Since components of the ESCRT machinery, particularly ALIX, are involved in endosomal sorting to the lysosome (*Dores et al., 2016*; *Murrow et al., 2015*), we asked whether eHAV trafficking to the lysosome is dependent on ESCRT. Strikingly, quasi-enveloped eHAV virions failed to reach the lysosome in cells depleted of ALIX, but not the ESCRT-III proteins CHMP1B or CHMP2A (*Figure 2E*, *Figure 2—figure supplement 5*). The apparent lack of a requirement for these ESCRT-III proteins could reflect less robust depletion of the targeted mRNA than that achieved with ALIX (*Figure 2—figure supplement 5*), or possibly the existence of functionally redundant homologs such as CHMP2B. Consistent with the imaging studies, depletion of ALIX had a strong negative effect on the early replication of eHAV, but not naked HAV (*Figure 2F*). Altogether, these results

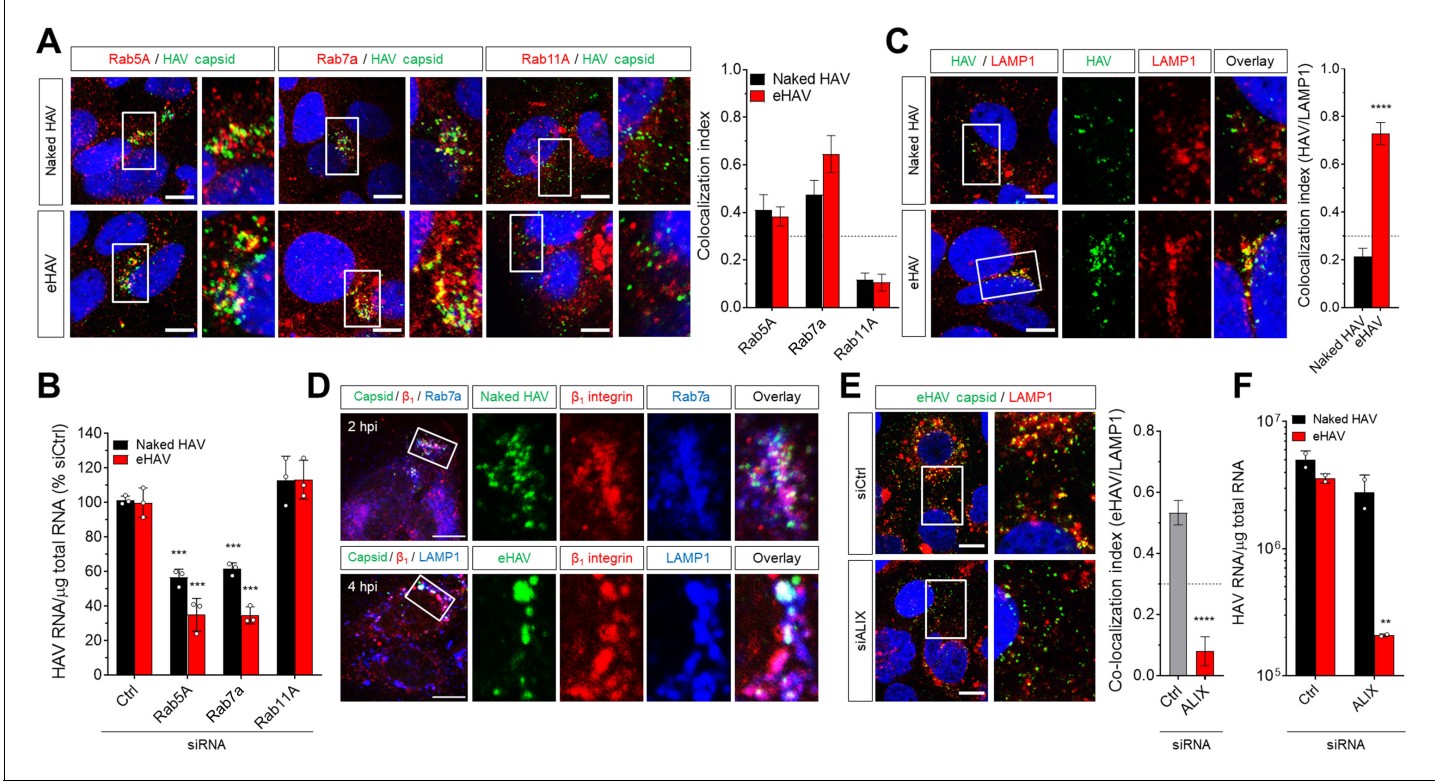

**Figure 2.** Distinct endocytic sorting of naked and quasi-enveloped HAV. (A) Confocal micrographs of Huh-7.5 cells immunolabeled with anti-HAV capsid (K24F2) and anti-Rab5A, Rab7a, or Rab11A at two hpi. Scale bar, 10 μm. (B) Effect of siRNA-mediated depletion of Rab GTPases on HAV and eHAV entry (mean ± SD, n = 3 independent experiments). See *Figure 2—figure supplement 1* for knockdown efficiencies. (C) Confocal micrographs of Huh-7.5 cells immunolabeled with anti-HAV capsid (K24F2) and anti-LAMP1 at six hpi. Scale bar, 10 μm. (D) Confocal micrographs of Huh-7.5 cells adsorbed with naked HAV or eHAV and immunolabeled with antibodies against HAV capsid (K24F2), β₁ integrin, and either Rab7 or LAMP1. Scale bar, 10 μm. (E) Confocal micrographs of Huh-7.5 cells previously transfected with a control or ALIX-specific siRNAs and immunolabeled with anti-HAV capsid (K24F2) and anti-LAMP1 at 12 hpi with eHAV. Scale bar, 10 μm. See *Figure 2—figure supplement 5* for knockdown efficiencies. (F) Effect of ALIX depletion by siRNA on eHAV and HAV entry and replication at 22 hpi (mean ± SD, n = 2 biological replicates for a representative experiment). For numeric data plotted in graphs associated with this figure, see *Figure 2—source data 1*.

DOI: https://doi.org/10.7554/eLife.43983.009

The following source data and figure supplements are available for figure 2:

**Source data 1.** Source data corresponding to *Figure 2*.
DOI: https://doi.org/10.7554/eLife.43983.015

**Figure supplement 1.** Immunoblots showing knockdown efficiency of siRNA transfections related to *Figure 2B*.
DOI: https://doi.org/10.7554/eLife.43983.010

**Figure supplement 2.** Huh-7.5 cells were inoculated with gradient-purified HAV or eHAV, fixed 6 hr post-adsorption, and subjected to immunostaining using antibodies against HAV capsid (K24F2) and the lysosomal protein VAMP8.
DOI: https://doi.org/10.7554/eLife.43983.011

**Figure supplement 3.** Huh-7.5 cells were inoculated with eHAV, fixed two hpi, and subjected to immunostaining using antibodies against HAV capsid (K24F2), β₁ integrin, and Rab7.
DOI: https://doi.org/10.7554/eLife.43983.012

**Figure supplement 4.** Huh-7.5 cells were pre-treated with the Alexa594-conjugated activating anti-β₁ integrin mAb -TS2/16 or IgG control for 20 min on ice to label cell surface β₁ integrin prior to virus adsorption.
DOI: https://doi.org/10.7554/eLife.43983.013

**Figure supplement 5.** Huh-7.5 cells were transfected with the indicated siRNAs for 72 hr prior to adsorption with eHAV.
DOI: https://doi.org/10.7554/eLife.43983.014

demonstrate that while both types of virions reach the late endosome, only eHAV is trafficked to the lysosome through an ALIX-dependent mechanism.

## Loss of the eHAV quasi-envelope occurs within the lysosome

Although an essential receptor molecule has yet to be identified for HAV (*Das et al., 2017*), studies with other picornaviruses (*Strauss et al., 2015*) suggest that the entry of both naked and quasi-enveloped virions is likely to involve binding of the capsid to a specific receptor that triggers uncoating. With quasi-enveloped eHAV, however, this can only occur after the membrane cloaking the capsid is degraded or fuses with a cellular membrane. Fusion seems unlikely given the absence of any virus-encoded proteins in the quasi-envelope (*McKnight et al., 2017*), whereas the selective targeting of eHAV to lysosomes suggests that the quasi-envelope, despite being stable at pH 5.0, might be degraded by cholesterol transporter proteins and hydrolytic enzymes expressed within late endosomes and lysosomes (*Feng et al., 2013*; *Kolter and Sandhoff, 2010*). A similar process has been suggested recently to facilitate the entry of phylogenetically-distinct, quasi-enveloped hepeviruses (*Yin et al., 2016*). Consistent with this hypothesis, partial siRNA-mediated depletion of the cholesterol transporter Niemann-Pick disease type C1 (NPC1) protein and lysosomal acid lipase (LAL), but not LAMP1, significantly impaired eHAV but not naked HAV infection, likely through altering the kinetics of quasi-envelope degradation (*Figure 3A*, *Figure 3—figure supplement 1*). Pharmacological inhibition of NPC1 and LAL with U18666A and Lalistat-2 (*Lu et al., 2015*; *Rosenbaum et al., 2010*), respectively, recapitulated these effects individually, and demonstrated an additive effect when combined (*Figure 3B*).

To confirm that the quasi-envelope is degraded within the lysosome, we harvested eHAV from supernatant fluids of infected Huh-7.5 cells and labelled the virions with the membrane-intercalating, red fluorescent dye PKH26 (*Figure 3C*). PKH26 irreversibly stains membrane lipids, allowing the labelled virions to be purified subsequently by isopycnic ultracentrifugation, and the fate of the quasi-envelope tracked by confocal microscopy following uptake into cells. We combined this approach with immunostaining cells with a monoclonal antibody to the capsid (K24F2) under minimal permeabilization conditions. This allowed us to visualize capsid antigen associated with PKH26-labeled membranes, and to differentiate eHAV from other EVs with similar density that co-purify in iodixanol gradients (*Feng et al., 2013*; *McKnight et al., 2017*). Confocal microscopy of Huh-7.5 cells inoculated with the gradient-purified, PKH26-labeled eHAV showed that capsid antigen was surrounded by PKH26-labeled membranes within lysosomes at ~6 hpi (*Figure 3D*). Thus, eHAV reaches the lysosome cloaked in membranes. At later time points, however, the PKH26 fluorescence was absent although the capsid antigen was still detected within lysosomes, consistent with the quasi-envelope being degraded within the lysosome to produce a naked capsid. As HAV replicates slowly in cell culture (*Whetter et al., 1994*), newly synthesized capsid antigen associated with the generation of progeny virions was not detected until after ~18–24 hpi, at which time there was minimal localization of capsid antigen within lysosomes.

As the data presented above suggest that LAL is important for eHAV entry (*Figure 3A,B*), we monitored loss of the eHAV membrane over time by following the decay of PKH26 fluorescence in cells infected in the presence or absence of the LAL inhibitor, Lalistat-2 (*Figure 3E*). As expected, inhibition of LAL delayed the kinetics of eHAV membrane loss without altering its trafficking to the lysosome, further supporting a model in which degradation of the eHAV membrane is facilitated by lysosomal enzymes. Interestingly, similar experiments using PKH26-labeled EVs collected from supernatant fluids of uninfected Huh-7.5 cells showed targeting of the PKH26 dye to $CD63^+$ endosomes, not lysosomes, without any decay in the fluorescence signal even as late as 24 hr post-inoculation (*Figure 3D*, *Figure 3—figure supplement 2*). This suggests that there may be specific targeting signals present within the eHAV membrane that are absent in unrelated EVs.

Although the quasi-envelope protects the virus against neutralization in quantal infectious focus-reduction neutralization assays, our previous studies show that eHAV (but not naked HAV virions) can be neutralized within an endocytic compartment when neutralizing IgG or IgA antibodies are added to cells as late as 4–6 hr after adsorption of the virus (*Feng et al., 2013*). To determine whether such neutralization is dependent upon LAL-mediated degradation of the quasi-envelope within lysosomes, we pre-treated Huh-7.5 cells with Lalistat-2 prior to infection with naked HAV or eHAV, and added anti-HAV-positive human plasma ('JC' plasma) at intervals following removal of the inoculum. Consistent with our previous results (*Feng et al., 2013*), neutralizing antibodies had no effect on replication of the nonenveloped, naked HAV under these conditions (*Figure 3F*, top), whereas replication of quasi-enveloped eHAV was substantially reduced when antibody was added

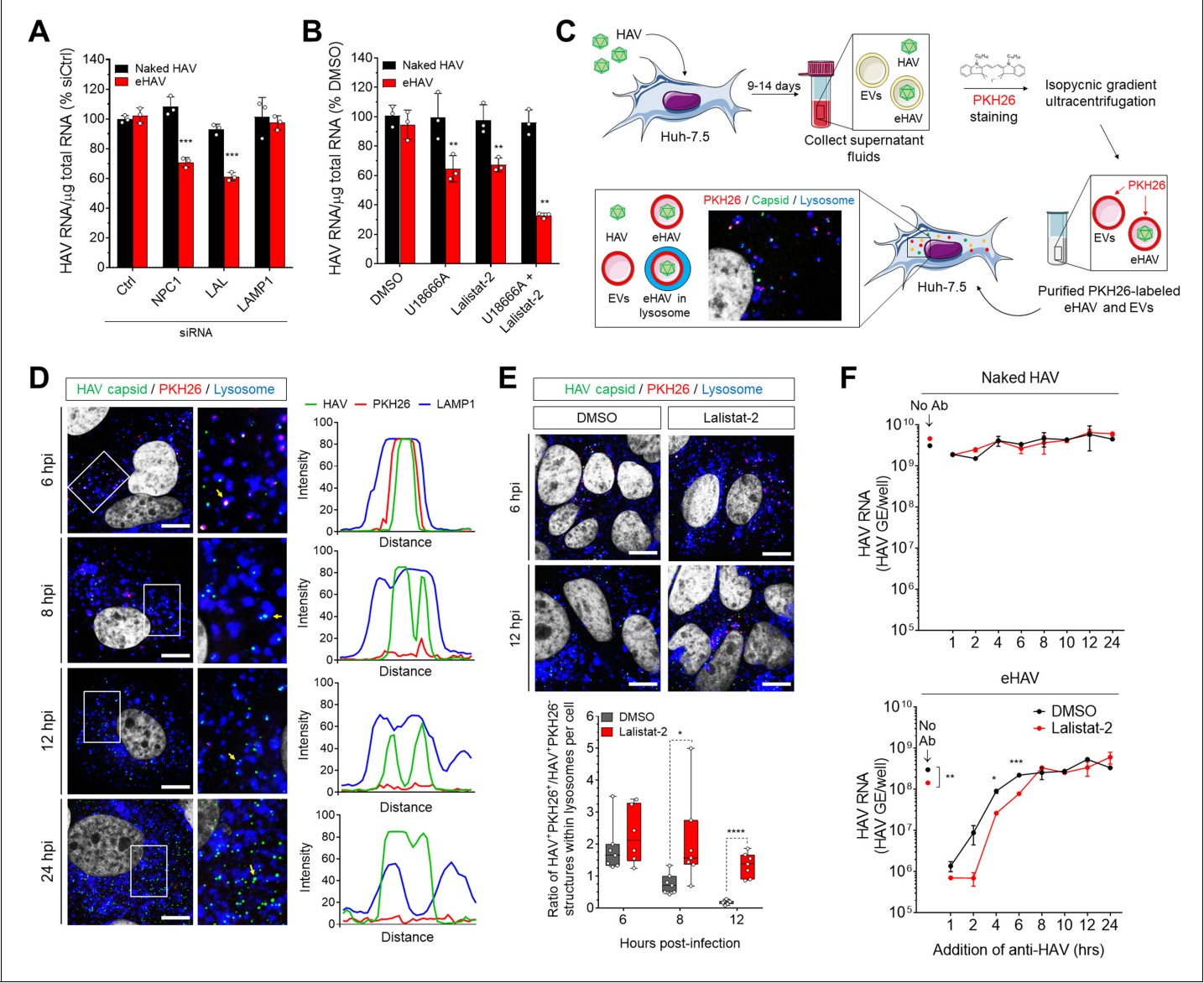

**Figure 3.** Degradation of the eHAV quasi-envelope occurs within the lysosome. (**A**) Effect of siRNA-mediated depletion of lysosomal-associated proteins on HAV and eHAV entry (mean ± SD, n = 3 independent experiments). See *Figure 3—figure supplement 1* for knockdown efficiencies. (**B**) Effect of inhibition of NPC1 or LAL by U18666A or Lalistat-2, respectively, on HAV and eHAV entry (mean ± SD, n = 3 independent experiments). (**C**) Strategy for membrane labeling of exosomes and eHAV with PKH26 dye. (**D**) Confocal micrographs of Huh-7.5 cells inoculated with PKH26-labeled eHAV and immunostained with anti-HAV capsid (K24F2) and anti-LAMP1. Histograms depict pixel intensities for the structures indicated with a yellow arrow. Scale bar, 10 μm. (**E**) Confocal micrographs of Huh-7.5 cells inoculated with PKH26-labeled eHAV in presence of Lalistat-2 and immunostained with anti-HAV capsid (K24F2) and anti-LAMP1. Graph displays the ratio of structures corresponding to PKH26-associated HAV capsid (green + red signal) over HAV capsids devoid of PKH26 fluorescence (green alone) within lysosomes per cell. Scale bar, 10 μm (**F**) Effect of Lalistat-2 on post-endocytic antibody-mediated neutralization of eHAV. Neutralizing anti-HAV (JC antibody) was added at the indicated intervals after removal of the inoculum and intracellular viral RNA was quantified at 48 hr (mean ±SD, n = 2 biological replicates for a representative experiment). For numeric data plotted in graphs associated with this figure, see *Figure 3—source data 1*.

DOI: https://doi.org/10.7554/eLife.43983.016

The following source data and figure supplements are available for figure 3:

**Source data 1.** Source data corresponding to *Figure 3*.
DOI: https://doi.org/10.7554/eLife.43983.019

**Figure supplement 1.** Immunoblots showing knockdown efficiency of siRNA transfections related to *Figure 3A*.
DOI: https://doi.org/10.7554/eLife.43983.017

*Figure 3 continued on next page*

*Figure 3 continued*

**Figure supplement 2.** Huh-7.5 cells were adsorbed with PKH26-labeled gradient-purified extracellular vesicles (EVs, fractions 9 to 11 in iodixanol gradient) obtained from supernatant fluids of uninfected Huh-7.5 cells, inocula was removed after 1 hr, and cells were fixed and immunostained with anti-CD63 at 8 or 24 hr post-inoculation.

DOI: https://doi.org/10.7554/eLife.43983.018

as late as ~4 hr after adsorption (*Figure 3F*, bottom). Importantly, however, the period of time during which eHAV was vulnerable to neutralization was extended significantly in cells treated with Lalistat-2 (*Figure 3F*, bottom). These results are consistent with Lalistat-2 slowing the transition of eHAV to a neutralization-susceptible state. Collectively, these data show that the quasi-enveloped eHAV capsid remains wrapped in membranes until the virion reaches the lysosome, where the quasi-envelope is degraded by lysosomal enzymes and cholesterol transporter proteins, rendering the capsid susceptible to antibody-mediated neutralization and, presumably, interactions with a yet-to-be-identified receptor.

## Kinetic and spatial differences in uncoating of HAV and eHAV capsids

To determine whether there are differences in the kinetics of uncoating of naked and quasi-enveloped capsids, we dually immunostained infected cells with a murine monoclonal antibody (K34C8) that recognizes an epitope expressed only on fully assembled capsids, and polyclonal human antibody (JC plasma) that recognizes both assembled capsids and assembly intermediates (14S pentamers) (*González-López et al., 2018*; *Stapleton et al., 1993*). Infections were done in the presence of cycloheximide to prevent synthesis of new viral proteins, such that uncoating would lead to loss of K34C8, but not JC, antigenicity. Confocal imaging of cells inoculated with naked HAV particles showed that the K34C8 signal was lost ~1–2 hpi without the capsid ever reaching the lysosome (*Figure 4A*). In contrast, K34C8-labeled capsid antigen was readily detected at ~4 hpi within lysosomes in cells infected with eHAV. This was followed by a progressive loss of the K34C8 signal, while JC antibody continued to detect capsid antigen within lysosomes up to 12 hpi. Thus, naked virions uncoat relatively rapidly upon entry, likely in a late endosomal compartment, whereas the capsids enclosed within eHAV vesicles do not uncoat until the virus reached the lysosome 4 hr or more following adsorbtion.

To assess how differences in the kinetics of uncoating of naked versus quasi-enveloped virions influence the onset and rate of polyprotein translation and viral RNA replication, we inoculated H1-HeLa cells with gradient-purified virions produced by a recombinant reporter virus that expresses nanoluciferase (HAV-NLuc) from within its polyprotein. Cells were infected in the presence or absence of the picornaviral RNA synthesis inhibitor, guanidine hydrochloride (GnHCl), and production of nanoluciferase measured over time (*Figure 4B*). Nanoluciferase expression resulting from translation of the incoming naked virus genome could be detected as early as ~4 hpi, while translation of the eHAV genome was not detectable until ~8 hpi. In both cases, nanoluciferase expression increased similarly in the presence or absence of GnHCl for ~10 hr once translation had commenced, following which accelerating increases in the absence of GnHCl indicated the production of new viral transcripts (*Figure 4B*). Thus, translation of the genomic RNA as well as the first round of RNA replication occurs sooner with naked HAV than with quasi-enveloped eHAV, consistent with the relatively rapid uncoating of the naked virion.

## Endolysosomal membrane damage and hepatovirus entry

The data presented above indicate that late endosomes and lysosomes are the terminal trafficking destinations of naked and quasi-enveloped HAV virions, respectively, and that the capsids associated with these different virion types uncoat and release their RNA genomes within these distinct endolysosomal compartments. However, it is not clear in either case how the RNA genome released from the capsid is then translocated from the endolysosomal lumen to the cytosol where it is translated on ribosomes. The VP4 capsid peptide possesses membrane pore forming activity in vitro (*Shukla et al., 2014*), but neither HAV nor eHAV has been shown previously to disrupt the integrity of endolysosomal membranes during infection. To determine whether hepatoviruses induce pores in endolysosomal membranes during entry as observed with other picornaviruses, we inoculated cells with eHAV or HAV in the presence of α-sarcin or restrictocin A, membrane-impermeable ribotoxins

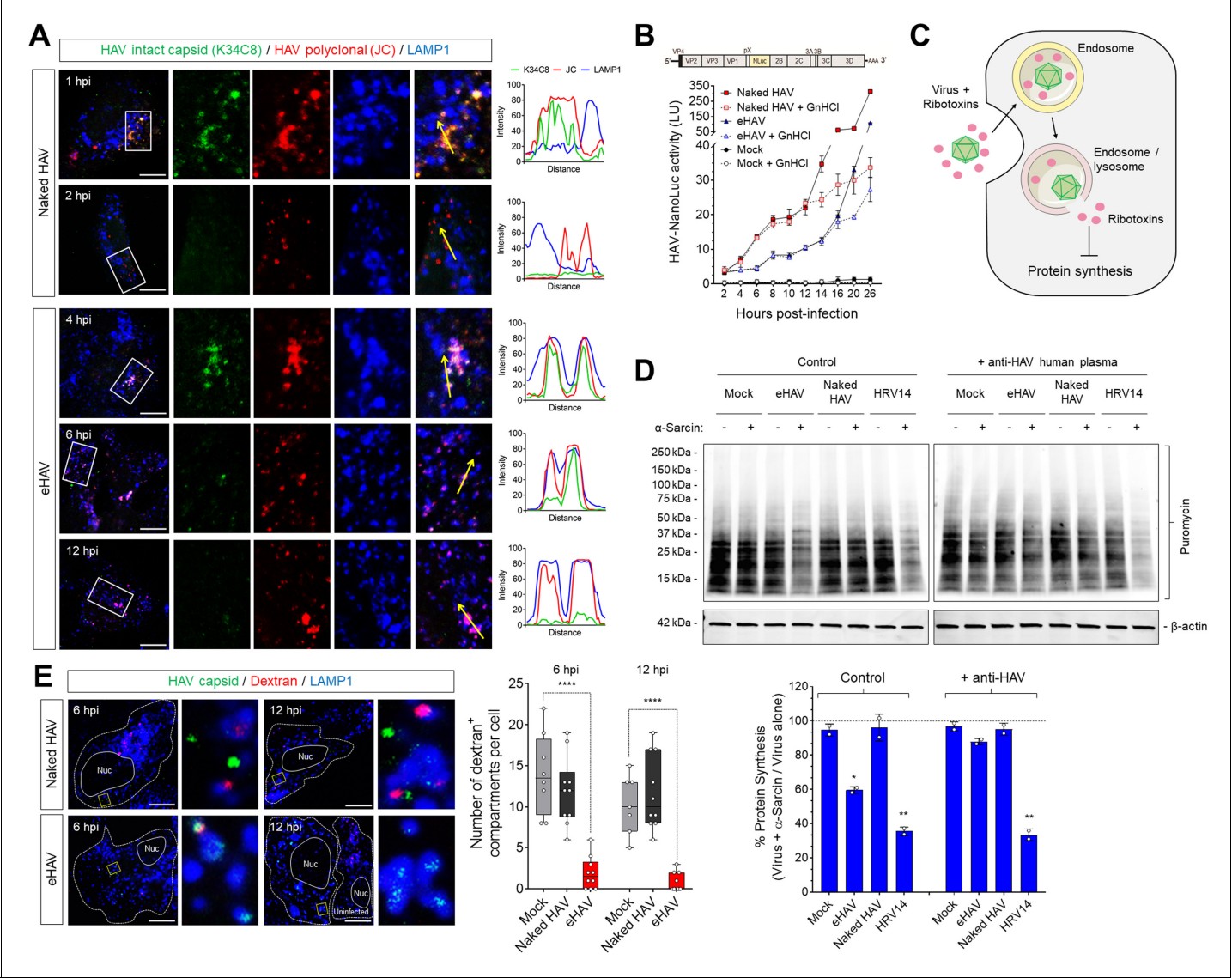

**Figure 4.** Different uncoating mechanisms for naked and quasi-enveloped HAV. (**A**) Confocal micrographs of Huh-7.5 cells adsorbed with HAV or eHAV in presence of cycloheximide immunostained with an antibody that recognizes intact HAV capsids (K34C8), a polyclonal human anti-HAV that recognizes denatured P1 proteins (JC), and LAMP1. Histograms depict pixel intensities for the structures indicated with a yellow arrow. Scale bar, 10 μm. (**B**) Kinetics of HAV RNA translation measured by nanoluciferase production by HAV-NLuc in H1-HeLa cells pre-treated or not with GnHCl to inhibit viral replication. (**C**) Schematic of the endolysosomal permeabilization assay using endocytosed ribotoxins. (**D**) Virus-induced endolysosomal membrane damage at six hpi. H1-HeLa cells were pulsed for 20 min with puromycin prior to cell lysis, and protein synthesis was analyzed by immunoblotting using an anti-puromycin antibody. Alternatively, viruses were incubated overnight at 4°C with human 'JC' plasma containing neutralizing HAV antibodies prior to adsorption and adsorbed as a mix. Band intensities were normalized to actin and protein synthesis is expressed as band intensity in cells infected in presence of α-sarcin relative to cells infected with virus alone (mean ± SD from two independent cultures of a representative of four independent experiments). (**E**) Micrographs of Huh-7.5 cells with lysosomes pre-loaded with fluorescently-conjugated dextran (10 kDa) prior to adsorption with HAV or eHAV and immunostained using antibodies against HAV capsid (K24F2) and LAMP1. Graphs display the number of dextran +compartments per cell (*n* = 7–10 cells per condition for a representative of three independent experiments). Scale bar, 10 μm. For numeric data plotted in graphs associated with this figure, see *Figure 4—source data 1*.

DOI: https://doi.org/10.7554/eLife.43983.020

The following source data and figure supplements are available for figure 4:

**Source data 1.** Source data corresponding to *Figure 4*.
DOI: https://doi.org/10.7554/eLife.43983.024

**Figure supplement 1.** Validation of puromycin incorporation into nascent proteins and Restrictocin A-induced inhibition of protein synthesis during viral entry.

*Figure 4 continued on next page*

*Figure 4 continued*

DOI: https://doi.org/10.7554/eLife.43983.021

**Figure supplement 2.** Kinetics of virus-induced endosomal membrane damage.

DOI: https://doi.org/10.7554/eLife.43983.022

**Figure supplement 3.** Lysosomes of Huh-7.5 cells were pre-loaded with fluorescently-conjugated dextran (10 kDa) overnight and cells were incubated with DMSO control or LLOMe for 2 hr prior to fixation and immunostaining with anti-LAMP1.

DOI: https://doi.org/10.7554/eLife.43983.023

that are released into the cytoplasm only if endolysosomal membranes are compromised (*Figure 4C*) (*Cuadras et al., 1997*; *Fernández-Puentes and Carrasco, 1980*; *Staring et al., 2017*). Global protein synthesis, quantified by puromycin incorporation, was significantly reduced in cells ~ 6 hr after adsorbtion of eHAV but not naked HAV, in the presence of either α-sarcin or restrictocin A (*Figure 4D*, *Figure 4—figure supplement 1A,B*). Reductions in protein synthesis were similar but not as strong as those observed in cells infected with human rhinovirus 14 (HRV14), included as a positive control in these experiments, and were not observed in cells inoculated with eHAV in the presence of neutralizing anti-HAV antibody which abrogated the ability of eHAV to induce endosomal escape of the ribotoxins (*Figure 4D*, *Figure 4—figure supplement 1B*). This effect was specific to eHAV, and naked HAV was never found to induce ribotoxin escape from endosomes at any time post-infection, even under conditions in which it was able to initiate translation of its genome (*Figure 4D*, *Figure 4—figure supplement 2A,B*).

To confirm that eHAV induces endolysosomal membrane injury and to determine more specifically that it occurs within lysosomes, as expected from the trafficking studies described above, we pre-loaded the lysosomes of Huh-7.5 cells with fluorophore-conjugated dextran prior to virus infection. Dextran is a complex, branched glucan that enters cells through fluid-phase pinocytosis and accumulates following its internalization in endolysosomal vesicles positive for LAMP1 and Rab7/LAMP1 (*Humphries et al., 2011*). The release of this pre-loaded dextran from lysosomes into the cytoplasm can be induced by lysosomotropic agents like L-leucyl-L-leucine methyl ester (LLOMe) (*Figure 4E*, *Figure 4—figure supplement 3*), providing a useful measure of lysosomal membrane permeability. As expected, cells infected with eHAV demonstrated significantly reduced numbers of dextran-positive compartments as early as six hpi (*Figure 4E*), a time at which eHAV was seen to be accumulating within lysosomes and sometimes co-localizing with dextran. In contrast, naked virions were never observed in lysosomes and did not alter the number of dextran-containing compartments, even at 12 hpi.

Collectively, these data indicate that eHAV uncoats within the lysosomal lumen and induces membrane damage congruent with release of its genome to the cytoplasm, whereas uncoating of naked HAV virions takes place within a late endosomal compartment in the absence of detectable endosomal membrane damage. Whether the absence of detectable endosomal rupture during HAV entry reflects a process that is mechanistically different from that by which eHAV releases its RNA genome across endolysosomal membranes is uncertain. It could be simply that late endosomal membranes breached by HAV are more capable of repair than the lysosomal membrane breached by eHAV.

## Role of PLA2G16 in eHAV and HAV entry and replication

PLA2G16 was identified recently as an essential entry factor for several members of the *Picornaviridae* (*Staring et al., 2017*). A phospholipase, it facilitates the safe translocation of the RNA genome from the endosome to the ribosome, providing for its escape from autophagosome-dependent degradation initiated by galectin-8 recruited to sites of endosomal membrane damage. To determine whether either HAV or eHAV entry is similarly dependent on PLA2G16, wild-type and CRISPR/Cas9-edited H1-Hela cells lacking expression of PLA2G16 (ΔPLA2G16 cells) or galectin-8 (ΔLGALS8 cells) (*Staring et al., 2017*) were infected with the nanoluc reporter virus. Surprisingly, neither *PLA2G16* or *LGALS8* knockout resulted in a difference in nanoluciferase expression 12 hpi with either HAV or eHAV (*Figure 5A*). Thus, unlike enteroviruses and cardioviruses (*Staring et al., 2017*), PLA2G16 is not required for safe transport of the hepatovirus genome from the endosomal lumen to ribosomes to initiate viral protein synthesis. Although *PLA2G16* knockout reduces the permissiveness of H1-HeLa cells for enterovirus infection (*Staring et al., 2017*), longer term studies demonstrated that the replication of both HAV and eHAV was enhanced in ΔPLA2G16 cells, with increased hepatovirus

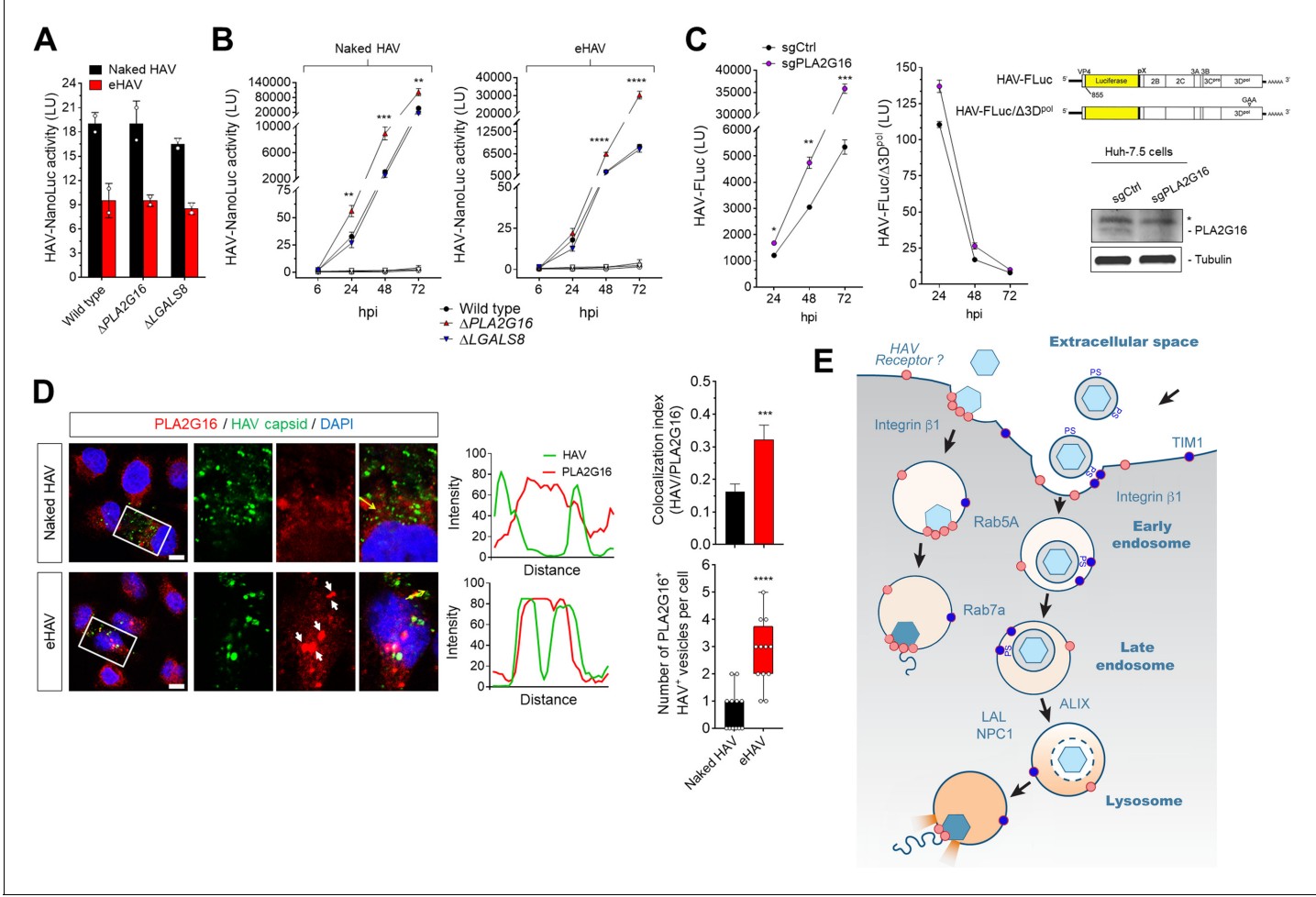

**Figure 5.** Dispensability of PLA2G16 during HAV entry. (**A**) Translation of HAV RNA during viral entry (~2000 GEs/cell, 12 hpi) measured by HAV-NLuc activity in H1-HeLa cells knocked-out for PLA2G16 or LGALS8 (mean ± SD, n = 2 biological replicates for a representative experiment). (**B**) Replication kinetics of naked or quasi-enveloped HAV/NLuc in H1-HeLa cells (mean ± SD, n = 3 biological replicates for a representative experiment). (**C**) Graphs showing luciferase activity expressed by transfected HAV-FLuc subgenomic replicon RNA (left) or a replication-incompetent mutant (right) in sgControl or sgPLA2G16 Huh-7.5 cells (mean ± SD, n = 2 biological replicates for a representative experiment). At the lower right is an immunoplot of PLA2G16 in these cells (*non-specific band). (**D**) Micrographs of H1-HeLa cells immunolabeled with anti-HAV capsid (K24F2) and anti-PLA2G16 at six hpi with naked HAV or eHAV. White arrows indicate sites of presumed membrane damage where PLA2G16 has accumulated. Scale bar, 10 μm. Histograms depict pixel intensities across the drawn arrow on the inset overlay. Graphs show the co-localization (Mander's coefficients) (n = 8 cells per condition) between HAV and PLA2G16 or the number of vesicles positive for both HAV capsid and PLA2G16. (**E**) Current model for cellular entry of naked and quasi-enveloped HAV virions. PS: phosphatidylserine. For numeric data plotted in graphs associated with this figure, see *Figure 5—source data 1*.

DOI: https://doi.org/10.7554/eLife.43983.025

The following source data and figure supplements are available for figure 5:

**Source data 1.** Source data corresponding to *Figure 5*.

DOI: https://doi.org/10.7554/eLife.43983.028

**Figure supplement 1.** Restriction of HAV replication by PLA2G16 in H1-HeLa cells.

DOI: https://doi.org/10.7554/eLife.43983.026

**Figure supplement 2.** Restriction of HAV replication by PLA2G16 in Huh-7.5 cells.

DOI: https://doi.org/10.7554/eLife.43983.027

RNA abundance, more dsRNA, and greater viral protein synthesis (*Figure 5B-Figure 5—figure supplement 1A,B*). The replication of naked HAV was similarly boosted in Huh-7.5 cells with siRNA- or CRISPR/Cas9-mediated depletion of PLA2G16 (*Figure 5B*, *Figure 4—figure supplement 2A,B*). Thus, PLA2G16 restricts, rather than promotes hepatovirus infection. Further experiments demonstrated that this restriction occurs at a post-entry step in replication of the HAV genome, as

replication of a subgenomic reporter replicon RNA (HAV-FLuc) lacking capsid-coding sequence was similarly enhanced in *PLA2G16* knockout Huh-7.5 cells (*Figure 5C*).

Although PLA2G16 is not required for entry of either virion type, confocal imaging showed co-localization of PLA2G16 with eHAV capsid antigen six hpi, presumably at sites of damaged lysosomal membranes (*Figure 5D*). Co-localization was not observed in cells infected with naked HAV. Thus, PLA2G16 appears to be recruited to sites of endolysosomal membrane damage induced by eHAV, behaving as it does in response to entry of other picornaviruses (*Staring et al., 2017*). However, PLA2G16 is not required to protect the RNA genome from autophagy during its delivery from the endolysosome to ribosomes, suggesting a fundamental difference in how hepatoviruses and other picornaviruses manage the final step in viral entry and deliver their RNA genomes across endolysosomal membranes.

## Discussion

Naked HAV and quasi-enveloped eHAV virions play distinct but equally important roles in the pathogenesis of hepatitis A, with naked HAV virions responsible for fecal-oral transmission of the virus between individuals, and quasi-enveloped eHAV mediating subsequent spread within the newly infected host (*Feng et al., 2013*; *Hirai-Yuki et al., 2016*). Here, we describe the entry pathways followed by these two virion types. The early endocytic trafficking routes for these different types of infectious virions are quite similar, but they are differentially sorted within the late endosome and uncoat their encapsidated RNA genomes in different endocytic compartments. The entry of both types of virions requires clathrin- and dynamin-dependent endocytosis and results in trafficking through Rab5+ and Rab7+ endosomal compartments, but only eHAV continues its trafficking to reach the lysosome, where degradation of the quasi-envelope and uncoating of the genome ensues (*Figure 5E*). In contrast, naked HAV uncoats in late endosomes shortly after internalization, resulting in relatively rapid translation of its genomic RNA. These results provide new insight into how quasi-enveloped hepatoviruses infect the cell, and are likely relevant to pathogenic quasi-enveloped viruses from other families, notably hepeviruses and noroviruses, that are released from infected cells in EVs of comparable size (*Nagashima et al., 2017*; *Santiana et al., 2018*).

The quasi-envelope represents an elegant strategy for evading antibody-mediated immune responses (*Feng et al., 2013*; *Takahashi et al., 2010*), but it imposes a need for additional steps in cellular entry prior to uncoating of the genome. PtdSer is displayed on the eHAV surface and the initial attachment of eHAV to cells occurs in part through the PtdSer receptor, TIM1 (HAVCR1) (*Das et al., 2017*; *Feng et al., 2015*). This interaction likely promotes virus spread within the liver, but TIM1 is not essential for infection (*Das et al., 2017*). We show here that integrin $\beta_1$ is also involved in quasi-enveloped virus entry. It is not required for attachment to the cell at 4°C, but is essential for efficient internalization of eHAV at 37°C (*Figure 1D,E*). It acts similarly in uptake of the naked virion, but must do so through interactions with a different ligand since integrin $\beta_1$ co-localized with both virion types on the cell surface, and within endosomes, hours before degradation of the quasi-envelope (*Figure 1I*). Consistent with this, activating antibodies that stabilize specific conformations of integrin $\beta_1$ (*Su et al., 2016*) differentially enhanced the uptake of the two virion types (*Figure 1H*). Integrin $\beta_1$ facilitates uptake of several other picornaviruses through interactions with their capsids (*Merilahti et al., 2012*), and the VP3 capsid protein of HAV contains conserved RGD and KGE integrin recognition motifs. However, neither motif is exposed on the surface of the naked HAV capsid (*Wang et al., 2015*), and thus both are unlikely ligands for integrin $\beta_1$. We conclude that integrin $\beta_1$ binds elsewhere on the HAV capsid, likely in association with an $\alpha$ integrin, and also interacts with a host protein on the surface of eHAV. We were unable to identify a specific $\alpha$ integrin involved in entry of either virion type, but sub-optimal knockdown conditions and/or promiscuity of integrin $\beta_1$ for an $\alpha$ integrin partner, may have masked a specific role for an $\alpha$ integrin(s).

Collectively, our data show trafficking of quasi-enveloped virus to the lysosome is essential for entry and uncoating of its genome. In addition to Rab5 and Rab7 GTPases, this requires the ESCRT accessory protein, ALIX (*Figure 2*), likely due to the critical role it plays in regulating trafficking from late endosomes to the lysosome (*Murrow et al., 2015*). ALIX mediates the ubiquitin-independent sorting and trafficking of certain G-protein coupled receptors (GPCRs) to lysosomes through interactions with YPX$_3$L motifs (*Dores et al., 2016*). The VP2 capsid protein of HAV possesses two such YPX$_3$L ALIX interaction motifs, and we have shown that these have an essential role in the biogenesis

of eHAV (*Feng et al., 2013*; *González-López et al., 2018*). Thus, ALIX is required for both efficient entry and release of quasi-enveloped hepatovirus. However, ALIX mediates sorting of eHAV to the lysosome prior to degradation of its membrane, and thus promotes the entry process at a point during which the VP2 YPX$_3$L motifs are occluded by the quasi-envelope, and moreover are within the lumen of the endosome and not available to interact with cytoplasmic ALIX. Rather than a direct interaction with the virus, the requirement for ALIX in eHAV entry is more likely to reflect a role for the ESCRT-associated protein in maturation and trafficking of the late endosome, akin to its suggested role in entry of human papillomavirus and arenaviruses (*Gräßel et al., 2016*; *Pasqual et al., 2011*).

Is there a specific signal that directs the endocytosed eHAV virion to lysosomes? We found that nonspecific EVs released from uninfected Huh-7.5 cells (most likely exosomes) did not traffic to lysosomes, and that PKH26-labelled membranes associated with these vesicles decayed within the cell much more slowly than the eHAV membrane (*Figure 3D*, *Figure 3—figure supplement 2*). These results are consistent with previous studies of exosome entry that show PKH26-labeled exosome membranes can be tracked within cells for 24 hr or longer after entry with little or no accumulation in lysosomes (*Dutta et al., 2014*; *Ringuette Goulet et al., 2018*; *Svensson et al., 2013*). This suggests the existence (or possibly the absence) of a specific targeting signal within the eHAV membrane that results in these virions being routed to the lysosomal lumen for degradation of the quasi-envelope. Quantitative proteomics studies provide some support for this hypothesis, as such studies show differential enrichment of host proteins associated with eHAV versus exosomes released from the same cells (*McKnight et al., 2017*). Within the lysosome, both LAL and NPC1 contribute to the degradation of the quasi-envelope required for uncoating of the genome (*Figure 3*), recapitulating the function of these lysosomal proteins in the quasi-enveloped hepevirus life cycle (*Yin et al., 2016*). No other picornavirus is known to be trafficked to the lysosome for uncoating.

What triggers uncoating of the eHAV capsid after degradation of the quasi-envelope within the lysosome, and is this trigger the same as that for naked capsids within late endosomes? For picornaviruses of the Aphthovirus genus, low pH alone is sufficient to promote the dissociation of the capsid into pentameric subunits (*Tuthill et al., 2009*). However, the HAV capsid is highly resistant to acid pH (*Siegl et al., 1981*). An alternative model is provided by poliovirus, which interacts with a specific receptor (CD155) that triggers a massive conformational rearrangement of the capsid providing for safe transfer of genomic RNA across the endosomal membrane (*Groppelli et al., 2017*; *Strauss et al., 2015*). This is accompanied by evidence of endosomal pore formation in ribotoxin assays, such as that we show here for quasi-enveloped hepatoviruses (*Figure 4D*) (*Schober et al., 1998*; *Staring et al., 2017*). However, we did not detect pore formation in cells infected with naked HAV, even under conditions in which we documented translation of the genomic RNA, and thus successful translocation of the genome across the endosomal membrane (*Figure 4—figure supplements 1* and *2*). Crystallographic studies have identified substantial differences in the structures of the hepatovirus and poliovirus capsids, including a domain swap in VP2 and the absence of a lipid 'pocket factor' in HAV (*Wang et al., 2015*). Interactions of the poliovirus capsid with its cellular receptor lead to the release of this pocket factor and an irreversible expansion of the capsid (*Strauss et al., 2015*). Whether a similar expansion of the HAV capsid occurs during the process of its uncoating is unknown. The HAV capsid is exceptionally stable, and how it uncoats is enigmatic (*Stuart et al., 2018*). Recent studies show that monoclonal antibodies that bind with high affinity are capable of destabilizing the HAV capsid, possibly mimicking a specific receptor interaction (*Wang et al., 2017*). The nature of that putative receptor remains unknown, but data we present here suggest that it is likely to be present in late endolysosomal membranes.

We found that the phospholipase, PLA2G16, is not required for safe translocation of the RNA genome in either virion type from the endolysosomal lumen to the cytoplasm. These observations stand in sharp contrast to the essential role of PLA2G16 in the entry of multiple other picornaviruses (*Staring et al., 2017*). Our data point collectively to fundamental differences in the mechanism(s) by which hepatoviruses and other picornaviruses accomplish the endgame in entry, delivering their RNA genome to the ribosome where synthesis of the viral polyprotein can commence. The data leave open the possibility that this process differs for eHAV and HAV not only in where it occurs spatially within the endolysosomal system, but also in its molecular details. This question is likely to be resolved only after it is determined whether a specific receptor protein exists that is capable of triggering uncoating of the hepatovirus capsid.

# Materials and methods

**Key resources table**

| Reagent type (species) | Designation | Source/reference | Identifiers | Additional information |
|---|---|---|---|---|
| Cell line (human) | Huh-7.5 | PMID: 12584342 | RRID: CVCL_7927 | Gift from Charles Rice |
| Cell line (human) | HeLa ΔPLA2G16 | PMID: 28077878 | | Gift from Thijn Brummelkamp |
| Cell line (human) | HeLa ΔGALS8 | PMID: 28077878 | | Gift from Thijn Brummelkamp |
| Biological sample (virus) | HM175/18 f hepatitis A virus (HAV) | PMID: 1705995 | GenBank KP879216.1 | |
| Biological sample (virus) | HM175/p16 hepatitis A virus (HAV) | PMID: 2833008 | GenBank KP879217.1 | |
| Biological sample (virus) | Human rhinovirus 14 (HRV14) | PMID: 8383233 | | Recovered from pWR3.26; gift from Roland Rueckert |
| Antibody (rabbit) | anti-clathrin heavy chain | Abcam | ab21679 | (WB) 1:2000, (IF) 1:1000 |
| Antibody (rabbit) | anti-AP2M1 | GeneTex | GTX113332 | (WB) 1:1000 |
| Antibody (rabbit) | anti-DNM2 | GeneTex | GTX113171 | (WB) 1:1000 |
| Antibody (rabbit) | anti-CAV1 | Abcam | ab2910 | (WB) 1:2000, (IF) 1:500 |
| Antibody (rabbit) | anti-FLOT1 | GeneTex | GTX104769 | (WB) 1:1000 |
| Antibody (rabbit) | anti-ARF6 | GeneTex | GTX112872 | (WB) 1:1000 |
| Antibody (rabbit) | anti-Rab5A | Abcam | ab18211 | (WB) 1:1000 |
| Antibody (rabbit, mAb) | anti-Rab7a | Cell Signaling | 9367 | (WB) 1:1000, (IF) 1:100 |
| Antibody (rabbit, mAb) | anti-Rab11A | Cell Signaling | 5589 | (WB) 1:1000, (IF) 1:100 |
| Antibody (rabbit, mAb) | anti-NPC1 | Abcam | ab134113 | (WB) 1:2000 |
| Antibody (rabbit) | anti-LAL | GeneTex | GTX101169 | (WB) 1:1000 |
| Antibody (mouse, mAb) | anti-ALIX | Santa Cruz | sc-53540 | (WB) 1:500 |
| Antibody (mouse, mAb) | anti-puromycin | Millipore | MABE343 | (WB) 1:10000 |
| Antibody (rabbit) | anti-PLA2G16 | Cayman Chemical | 10337 | (WB) 1:200 |
| Antibody (goat) | anti-Galectin-8 | R and D Systems | AF1305 | (WB) 1:500 |
| Antibody (mouse, mAb) | anti-tubulin | Sigma | T6199 | (WB) 1:20000 |
| Antibody (rabbit) | and anti-actin | Sigma | A2066 | (WB) 1:5000 |
| Antibody (sheep) | anti-VCAM-1 | R and D Systems | AF809 | (WB) 1:250 |
| Antibody (sheep) | anti-ICAM-1 | R and D Systems | AF720 | (WB) 1:250 |
| Antibody (rat, mAb) | anti-Tspan8 | R and D Systems | MAB4734 | (WB) 1:150 |
| Antibody (rabbit, mAb) | anti-integrin β1 | Cell Signaling | 9699 | (WB) 1:1000 |
| Antibody (rabbit, mAb) | anti-integrin β3 | Cell Signaling | 13166 | (WB) 1:500 |
| Antibody (rabbit, mAb) | anti-integrin α1 | R and D Systems | MAB5676 | (WB) 1:250 |
| Antibody (rabbit, mAb) | anti-integrin α2 | Abcam | ab133557 | (WB) 1:500 |
| Antibody (rabbit) | anti-integrin α3 | Millipore | AB1920 | (WB) 1:250 |
| Antibody (rabbit, mAb) | anti-integrin α4 | Cell Signaling | 8440 | (WB) 1:250 |
| Antibody (rabbit) | anti-integrin α5 | Cell Signaling | 4705 | (WB) 1:700 |
| Antibody (rabbit) | anti-integrin αV | Cell Signaling | 4711 | (WB) 1:700 |
| Antibody (rabbit) | anti-integrin α6 | GeneTex | GTX100565 | (WB) 1:500 |

*Continued on next page*

*Continued*

| Reagent type (species) | Designation | Source/reference | Identifiers | Additional information |
|---|---|---|---|---|
| Antibody (rabbit) | anti-integrin α7 | Thermo Fisher Sci | PA5-37435 | (WB) 1:250 |
| Antibody (rabbit) | anti-integrin α8 | Novus Biologicals | NBP1-59940 | (WB) 1:250 |
| Antibody (mouse, mAb) | anti-integrin α9 | R and D Systems | MAB4574 | (WB) 1:250 |
| Antibody (mouse, mAb) | anti-integrin β1 | Abcam | ab30394 | (IF) 1:100 |
| Antibody (rabbit, mAb) | anti-integrin α5 | Abcam | ab150361 | (IF) 1:250 |
| Antibody (rabbit, mAb) | anti-integrin αV | Abcam | ab179475 | (IF) 1:500 |
| Antibody (rabbit, mAb) | anti-Rab11 | Cell Signaling | 5589 | (IF) 1:100 |
| Antibody (rabbit, mAb) | anti-LAMP1 | Cell Signaling | 9091 | (IF) 1:200 |
| Antibody (rabbit, mAb) | anti-VAMP8 | Abcam | ab76021 | (IF) 1:250 |
| Antibody (mouse, mAb) | anti-CD63 | BD Biosciences | #556019 | (IF) 1:50 |
| Antibody (rabbit) | anti-PLA2G16 | Sigma | H8290 | (IF) 1:50 |
| Antibody (mouse, mAb) | anti-dsRNA | Scicons | J2 | (IF) 1:1000 |
| Antibody (human) | JC human anti-HAV | PMID: 23542590 | JC | (IF) 1:600 |
| Antibody (mouse, mAb) | K24F2 anti-HAV capsid | PMID: 6315771 | K24F2 | (IF) 1:100 |
| Antibody (mouse, mAb) | K34C8 anti-HAV capsid | MacGregor et al. | K34C8 | (IF) 1:300 |
| Antibody (mouse, mAb) | anti-integrin β1 K-20 | Santa Cruz | sc-18887 | 10 µg ml$^{-1}$ |
| Antibody (mouse, mAb) | anti-integrin β1 TS2/16 | Santa Cruz | sc-53711 | 10 µg ml$^{-1}$ |
| Antibody (mouse, mAb) | anti-integrin β1 8E3 | Millipore | MABT199 | 10 µg ml$^{-1}$ |
| Antibody (mouse, mAb) | anti-integrin β1 HUTS-2 | Millipore | MAB2079Z | 10 µg ml$^{-1}$ |
| Chemical compound | dynasore | Millipore | #324410 | 80 µM |
| Chemical compound | chlorpromazine | Sigma-Aldrich | #C8138 | 10 µg ml$^{-1}$ |
| Chemical compound | filipin | Sigma-Aldrich | #F9765 | 1 µg ml$^{-1}$ |
| Chemical compound | cytochalasin D | Sigma-Aldrich | #C2618 | 20 µM |
| Chemical compound | latrunculin A | Sigma-Aldrich | #428026 | 1 µM |
| Chemical compound | 5-(N-Ethyl-N-isopropyl)amiloride) | Sigma-Aldrich | #A3085 | 1 µM |
| Chemical compound | wortmannin | Sigma-Aldrich | #W1628 | 1 µM |
| Chemical compound | NSC23766 | Sigma-Aldrich | #SML0952 | 300 µM |
| Chemical compound | dynarrestin | PMID: 29396292 | | 25 µM, gift from Jared Sterneckert |
| Chemical compound | heparin | Sigma-Aldrich | #H3149 | 10 µg ml$^{-1}$ |
| Chemical compound | Lalistat-2 | PMID: 20557099 | | 200 µM, gift from Paul Helquist |
| Chemical compound | U18666A | Sigma-Aldrich | #U3633 | 2 µg ml$^{-1}$ |
| Chemical compound | guanidine hydrochloride | Sigma | #G3272 | 5 mM |
| Chemical compound | cycloheximide | Sigma | #C7698 | 25 µg ml$^{-1}$ |
| Chemical compound | puromycin | InvivoGen | CAS 58-58-2 | 5–20 µg ml$^{-1}$ |
| Chemical compound | α-sarcin | Santa Cruz | #sc-204427 | 100 µg ml$^{-1}$ |
| Chemical compound | Restrictocin A | Sigma-Aldrich | #R0389 | 50 µg ml$^{-1}$ |
| Chemical compound | L-Leucyl-L-Leucine-methyl ester | Cayman Chemical | #16008 | 2 mM |

## Cells

Huh-7.5 cells (obtained from Charles Rice, Rockefeller University) were maintained in Dulbecco's modified Eagle medium (DMEM) supplemented with 10% fetal bovine serum (FBS). Wild type, ΔPLA2G16, and ΔLGALS8 H1-HeLa cells have been previously characterized (*Staring et al., 2017*) and were maintained in DMEM supplemented with 10% FBS and 2 mM GlutaMAX. Cell lines were validated by phenotypic screening and confirmed to be mycoplasma-free using a PCR detection kit (Sigma-Millipore, # MP0035). To produce knockout Huh-7.5 cells, CRISPR/Cas9-expressing lentiviruses were generated by co-transfection of 293FT cells with sgRNA lentivectors (Applied Biological Materials, *Supplementary file 1*) and a third-generation lentivirus packaging mix kit (Applied Biological Materials, #LV053-G074). Supernatant fluids were collected at 48–72 hr post-transfection, spun to remove cell debris, and filtered through a 0.45 μm filter syringe. Lentivirus transduction of Huh-7.5 or H1-HeLa cells was performed by supplementation of 8 μg.ml$^{-1}$ polybrene followed by antibiotic selection with 6 or 10 μg.ml$^{-1}$ puromycin, respectively, was performed as previously described (*Das et al., 2017*). All cells were maintained at 37°C in a 5% $CO_2$ atmosphere.

## Preparation of purified HAV and eHAV

Infectious clones of HM175/p16.2 virus (low-passage, non-cytopathic, cell culture-adapted, GenBank KP879217.1 (*McKnight et al., 2017*)) and HM175/18.2 (high cell culture passage, rapid replication, cytopathic, GenBank KP879216.1 (*González-López et al., 2018*; *Zhang et al., 1995*) are variants of the HM175 strain (*Jansen et al., 1988*; *Taylor et al., 1993*) and have been previously described. The HM175/18 f.2-NanoLuc (HAV-NLuc) plasmid was created by PCR amplifying the NLuc ORF using pNL1.1 (Promega) as template and oligos containing the tri-glycine sequence flanked by *Xba*I and *Bam*H1 restriction sites. The PCR amplicon was enzymatically digested and ligated into digested pSK-2A-Zeo-2B plasmid. The resulting plasmid was further digested with *Sac*I/*Pfl*MI to release the entire 2A-NLuc-2B fragment which was then ligated into a similarly digested HM175/18 f parental plasmid. Infectious HAV mRNA transcripts were generated in vitro using the T7 RiboMAX$^{TM}$ Express Large-Scale RNA Production System (Promega) as per manufacturer's protocol and transfected into Huh-7.5 cells by electroporation in a Gene Pulser Xcell Total System (Bio-Rad) as previously described (*Feng et al., 2013*). Cell culture supernatant fluids were then collected (9 to 15 days post-transfection) and centrifuged at 1,000 ×$g$ for 10 min at 4°C to remove debris and further clarified at 10,000 ×$g$ for 30 min at 4°C. The virus was concentrated by ultracentrifugation at 100,000 ×$g$ for 60 min at 4°C, and the resulting pellet was resuspended in 250 μl phosphate buffer saline (PBS) and loaded on top of a five-step gradient of 8% to 40% iodixanol (OptiPrep, Sigma) and centrifuged at 165,915 ×$g$ (37,000 rpm) for 24 hr at 4°C in a Beckman SW55i rotor using a Beckman Optima LE-80K ultracentrifuge. Approximately 20 fractions were collected from the top of the isopycnic gradient, and HAV RNA content and density were quantified by reverse transcription-quantitative PCR (RT-qPCR) and refractometry, respectively, as previously described (*Feng et al., 2013*; *McKnight et al., 2017*). Fractions containing eHAV and HAV at the appropriate buoyant densities (for eHAV, approximately 1.08 g/cm$^3$, fractions 9 to 11; for naked HAV, approximately 1.22 g/cm$^3$, fractions 18–19) were stored in aliquots at –80°C until use.

## Firefly luciferase and NanoLuciferase assays

Huh-7.5 cells were seeded on 96-well clusters and transfected with in vitro transcribed subgenomic HAV-FLuc replicon or a replication-incompetent mutant (*González-López et al., 2018*; *Yi and Lemon, 2002*) using *Trans*IT-mRNA transfection kit (Mirus Bio, #MIR2250) according to the manufacturer's instructions. Cells were harvested in 1 × passive lysis buffer (PLB, Promega) at the indicated times post-transfection and luciferase activity was measured using a firefly luciferase assay system (Promega, #E1501). For HAV-NanoLuc assays, cells were lysed for 5–10 min in 1 × PLB and mixed with 1 × substrate for *Oplophorus* luciferase (NanoLight Technology, #325) according to the manufacturer's instructions. All luciferase readings were obtained on a BioTek Synergy two microplate reader.

## Pharmacological agents and functional antibodies

For viral entry assays, 5 × 10$^4$ Huh-7.5 cells seeded on 12-well chamber slides were pre-treated with the indicated inhibitors at the following concentrations: 80 μM dynasore (Millipore, #324410), 10 μg.

ml$^{-1}$ chlorpromazine (Sigma-Aldrich, #C8138), 1 µg.ml$^{-1}$ filipin (Sigma-Aldrich, #F9765), 20 µM cyto-chalasin D (Sigma-Aldrich, #C2618), 1 µM latrunculin A (Sigma-Aldrich, #428026), 1 µM EIPA (5-(N-Ethyl-N-isopropyl)amiloride) (Sigma-Aldrich, #A3085), 1 µM wortmannin (Sigma-Aldrich, #W1628), 300 µM NSC23766 (Sigma-Aldrich, #SML0952), 25 µM dynarrestin (kindly provided by Dr. Jared Sterneckert, Max Planck Institute for Molecular Biomedicine, (*Höing et al., 2018*)), 10 µg.ml$^{-1}$ hepa-rin (Sigma-Aldrich, #H3149), 200 µM Lalistat-2 (kindly provided by Paul Helquist and Bruce Mal-encon, University of Notre Dame), 2 µg.ml$^{-1}$ U18666A (Sigma-Aldrich, #U3633), or dimethyl sulfoxide (DMSO) solvent control in supplemented DMEM for 60–120 min at 37°C prior to virus adsorption. For functional integrin assays, cells were incubated with 100 µM RGD peptide (Santa Cruz Biotechnology, #sc-201176) for 2 hr; or 10 µg.ml$^{-1}$ of either mouse IgG (Abcam, #ab37355), K-20 (Santa Cruz Biotechnology, #sc-18887), TS2/16 (Santa Cruz Biotechnology, #sc-53711), 8E3 (Millipore-Sigma, #MABT199), or HUTS-4 (Millipore-Sigma, #MAB2079Z) for 20 min on ice prior to virus adsorption at 37°C. Other inhibitors include 5 mM guanidine hydrochloride (Sigma, #G3272), 25 µg.ml$^{-1}$ cycloheximide (Sigma, #C7698), 5–20 µg.ml$^{-1}$ puromycin (InvivoGen), 100 µg.ml$^{-1}$ α-sar-cin (Santa Cruz Biotechnology, #sc-204427), 50 µg.ml$^{-1}$ Restrictocin A (Sigma-Aldrich, #R0389), and 2 mM L-Leucyl-L-Leucine-methyl ester (Cayman Chemical, #16008).

## Quantitative (real-time) reverse transcription-PCR

RNA was extracted from cell lysates with the RNeasy Kit (Qiagen) and cDNA was synthesized with oligo(dT)$_{20}$ followed by RNaseH digestion. HAV RNA GEs were quantified in a SYBR Green Real-Time qPCR (Bio-Rad) assay against a synthetic RNA standard curve using primers targeting the HAV 5' untranslated region as previously described (*Feng et al., 2013*) and HAV RNA levels were normal-ized to total µg RNA. For siRNA-mediated knockdown efficiency, host mRNA target abundance was determined using gene-specific primers (*Supplementary file 1*) and normalized to glyceraldehyde-3-phosphate dehydrogenase levels; efficiency was calculated as the percent mRNA expression relative to non-targeting control siRNA samples.

## siRNA-mediated knockdowns

Huh-7.5 cells were transfected with 50–75 nM gene-specific SMARTPool ON-TARGETplus siRNAs (Dharmacon, *Supplementary file 1*) using the Lipofectamine RNAiMAX transfection reagent (Thermo Fisher Scientific) according to the manufacturer's instructions. Three-to-four days post-transfection, cells were adsorbed with equal quantities of HAV genome equivalents (GEs) of purified naked or quasi-enveloped HAV (~10 GEs per cell) for 1 hr at 37°C. The inoculum was removed, the cells were rinsed with PBS and incubated at 37°C in fresh medium.

## SDS-PAGE and immunoblotting

Cells were lysed in radioimmunoprecipitation assay (RIPA) buffer (50 mM Tris-HCl [pH 7.4], 1% NP-40, 0.25% sodium deoxycholate, 150 mM NaCl, 1 mM EDTA, 1% sodium dodecyl sulfate [SDS]) sup-plemented with a cocktail of protease and phosphatase inhibitors for 20 min on ice and then clarified at 14,000 ×$g$ for 10 min at 4°C. Total protein concentration was determined using a bicinchoninic acid assay (Thermo Fisher Scientific). A total of 5–20 µg of protein was boiled for 5 min in Laemmli sample buffer, resolved by SDS-polyacrylamide gel electrophoresis (SDS-PAGE), and transferred to a polyvinylidene fluoride (PVDF) membrane by standard methods. Membranes were blocked with Odyssey blocking buffer (LI-COR Bioscience), probed with the indicated primary antibodies, and incubated with infrared-conjugated (IRDye) secondary antibodies (LI-COR Biosciences). Proteins were visualized using an Odyssey Infrared Imaging System (LI-COR Biosciences).

## Indirect immunofluorescence

Huh-7.5 or H1-HeLa cells seeded on 8-well chamber slides were adsorbed with equal quantities of HAV genome equivalents (GEs) of purified naked or quasi-enveloped HAV (~1000 GEs per cell) for 1 hr at 37°C. The inoculum was removed, the cells were rinsed with PBS and reincubated at 37°C in fresh medium. Cells were then fixed in 4% paraformaldehyde (PFA) for 20 min at the indicated hours post-inoculation (hpi) and, unless stated otherwise, permeabilized with 0.25% Triton X-100 in PBS for 10 min. Slides were blocked with 5–10% normal goat serum (Sigma-Aldrich) in PBS for 60 min, and incubated with the indicated primary antibodies diluted in 0.1% IgG- and protease-free bovine

serum albumin (BSA) (Jackson ImmunoResearch, #001–000) for 1–2 hr at room temperature. Slides were extensively rinsed in PBS and incubated with the appropriate species-specific Alexa Fluor-conjugated secondary antibodies (Thermo Fisher Scientific) diluted in 0.1% IgG- and protease-free BSA for 1 hr at room temperature. Nuclei were counterstained with 300 nM DAPI (4',6-diamidino-2-phenylindole) and coverslips were mounted on slides using ProLong Gold (Thermo Fisher Scientific, #P36930).

## Antibodies

Antibodies used for immunoblotting and their corresponding dilutions were anti-clathrin heavy chain (Abcam, ab21679, 1:2000), anti-AP2M1 (GeneTex, GTX113332, 1:1000), anti-DNM2 (GeneTex, GTX113171, 1:1000), anti-CAV1 (Abcam, ab2910, 1:2000), anti-FLOT1 (GeneTex, GTX104769, 1:1000), anti-ARF6 (GeneTex, GTX112872, 1:1000), anti-Rab5A (Abcam, ab18211, 1:1000), anti-Rab7a (Cell Signaling, 9367, 1:1000), anti-Rab11 (Cell Signaling, 5589, 1:1000), anti-NPC1 (Abcam, ab134113, 1:2000), anti-LAL (GeneTex, GTX101169, 1:1000), anti-ALIX (Santa Cruz Biotech, sc-53540, 1:500), anti-puromycin (Millipore, MABE343, 1:10000), anti-PLA2G16 (Cayman Chemical, 10337, 1:200), anti-Galectin-8 (R and D Systems, AF1305, 1:500), anti-tubulin (Sigma, T6199, 1:20000), and anti-actin (Sigma, A2066, 1:5000), anti-VCAM-1 (R and D Systems, AF809, 1:250), anti-ICAM-1 (R and D Systems, AF720, 1:250), anti-Tspan8 (R and D Systems, MAB4734, 1:150), anti-integrin β1 (Cell Signaling, 9699, 1:1000), anti-integrin β3 (Cell Signaling, 13166, 1:500), anti-integrin α1 (R and D Systems, MAB5676, 1:250), anti-integrin α2 (Abcam, ab133557, 1:500), anti-integrin α3 (Millipore, AB1920, 1:250), anti-integrin α4 (Cell Signaling, 8440, 1:250), anti-integrin α5 (Cell Signaling, 4705, 1:700), anti-integrin αV (Cell Signaling, 4711, 1:700). anti-integrin α6 (GeneTex, GTX100565, 1:500), anti-integrin α7 (Thermo Fisher Sci, PA5-37435, 1:250), anti-integrin α8 (Novus Biologicals, NBP1-59940, 1:250), and anti-integrin α9 (R and D Systems, MAB4574, 1:250). Antibodies user for indirect immunofluorescence and their corresponding dilutions were anti-clathrin heavy chain (Abcam, ab21679, 1:1000), anti-CAV1 (Abcam, ab2910, 1:500), anti-integrin β1 (Abcam, ab30394, 1:100), anti-integrin α5 (Abcam, ab150361, 1:250), anti-integrin αV (Abcam, ab179475, 1:500), anti-Rab5A (Cell Signaling, 3547, 1:200), anti-Rab7a (Cell Signaling, 9367, 1:100), anti-Rab11 (Cell Signaling, 5589, 1:100), anti-LAMP1 (Cell Signaling, 9091, 1:200), anti-VAMP8 (Abcam, ab76021, 1:250), anti-CD63 (BD Biosciences, #556019, 1:50), anti-PLA2G16 (Sigma, H8290, 1:50), J2 anti-dsRNA (Scicons, J2 clone, 1:1,000), postconvalescent polyclonal anti-HAV human plasma 'JC' (*Feng et al., 2013*, 1:600), anti-HAV capsid K24F2 and K34C8 (*MacGregor et al., 1983*, 1:100 and 1:300, respectively).

## Transferrin and cholera toxin B uptake assays

Huh-7.5 cells seeded on 8-well chamber slides were treated with DMEM supplemented with the indicated inhibitors for 1 hr. Cells were then rinsed, placed on ice for 10 min, and incubated with 10–25 $\mu g.ml^{-1}$ Alexa 594-conjugated Transferrin (Thermo Fisher, #T13343) or cholera toxin subunit B (Thermo Fisher, #34777) diluted in supplemented DMEM for 15–20 min at 37°C. Cells were then fixed with 4% paraformaldehyde (PFA) and nuclei were counterstained with 300 nM DAPI.

## PKH26 staining of EVs and eHAV membranes

Supernatant fluids from uninfected or HAV-infected Huh-7.5 cells were clarified and concentrated by ultracentrifugation at 100,000 $\times g$ for 1 hr as described above. The pellet was then resuspended 250 µl of Diluent C and mixed with 2 µM PKH26 red fluorescent cell linker (Sigma-Aldrich, MIDI26) diluted in Diluent C for 5 min according to the manufacturer's instructions. The staining was blocked with 500 µl FBS for 3 min and the labeled vesicles/eHAV were loaded onto an iodixanol gradient and ultracentrifuged at 165,915 $\times g$ (37,000 rpm) for 24 hr at 4°C as described above. Fractions containing eHAV and EVs at the appropriate buoyant densities (approximately 1.08 g/cm$^3$, fractions 9 to 11) were stored in aliquots at 4°C until use. Recipient Huh-7.5 cells were inoculated with a 1:10 dilution of the fractions diluted in complete DMEM and fixed in 4% PFA. Slides were blocked in 10% normal goat serum, and incubated simultaneously with anti-HAV capsid (K24F2) and anti-LAMP1 for 1 hr diluted in 0.01% saponin (Sigma-Aldrich, S2149), carefully rinsed, and incubated with a mix of DAPI and Alexa fluor-conjugated secondary antibodies diluted in 0.01% saponin for 45 min at room temperature. Slides were mounted on ProLong Gold.

## Antibody-mediated HAV neutralization assay

Huh-7.5 cells seeded on 12-well clusters ($1 \times 10^5$ cells per well) were pre-treated with 200 μM Lalistat-2 or DMSO solvent control for 1 hs at 37°C and then adsorbed with equal quantities of HAV genome equivalents (GEs) of gradient-purified naked or quasi-enveloped HAV (~1 GE per cell) for 1 hr at 37°C. The inoculum was removed, cells were rinsed three times with PBS, and replaced with fresh DMEM supplemented with 10% FBS and 200 μM Lalistat-2 or DMSO. At the indicated times post-infection, media was replaced with postconvalescent human plasma ('JC plasma') collected several months following symptomatic acute hepatitis A (*Feng et al., 2013*) or normal human serum control diluted 1:50 in DMEM. Intracellular viral RNA was harvested at 48 hpi, cDNA was synthesized, and HAV RNA levels were quantified by RT-qPCR as described above.

## Endosomal and lysosomal membrane integrity assays

For endosomal membrane integrity, H1-HeLa cells were inoculated with equal amounts (~1000 HAV GEs per cell) of gradient-purified HAV or eHAV, or 10 plaque-forming units (PFU) per cell of human rhinovirus-14 (HRV-14) (*McKnight and Lemon, 1996*) in supplemented DMEM presence of α-sarcin or Restrictocin A previously reconstituted in sterile ultrapure water and incubated for 6 hr at 37°C (for HAV) or 33°C (for HRV-14). Cells were then incubated with supplemented DMEM containing 20 μg.ml$^{-1}$ puromycin for 20 min, rinsed twice with PBS, and total protein lysates were harvested as described above. Specific puromycin incorporation was validated by pre-treating cells with cycloheximide for 30 min prior to the puromycin pulse (*Figure 4—figure supplement 1*). For lysosomal membrane integrity analysis, Huh-7.5 cells seeded in 8-well chamber slides were loaded with 100 μg.ml$^{-1}$ anionic-lysine fixable Alexa Fluor 594-conjugated dextran (10 kDa) (Thermo Fisher Scientific, #D22913) diluted in supplemented DMEM for 16 hr at 37°C. Cells were rinsed with PBS, inoculated with purified HAV or eHAV (1000 HAV GEs per cell), and fixed as indicated.

## Confocal microscopy and image analyses

Slides were examined with an Olympus FV10000 laser-scanning confocal microscope equipped with a super corrected 60×/1.4 NA oil-immersion objective and a dichroic mirror DM405/488/543/635 was used for all experiments. The pinhole was maintained at 1 Airy unit and images were acquired in two separate channels to prevent bleed-through. The excitation/emission wavelengths were 405 nm/425–520 nm for DAPI, 488 nm/500–520 nm for Alexa Fluor 488, 543 nm/555–647 nm for Alexa Fluor 594 or PKH26, and 635 nm/647–700 nm for Alexa Fluor 647. Intensity plot profiles were generated using the ImageJ software and co-localization indexes (Mander's coefficients) were obtained with the Just Another Colocalisation Plugin (JACoP) module for ImageJ. All micrographs are representative of at least 10 images for each sample per experiment, and each experiment was performed at least twice. Images were processed for presentation using Photoshop CS4.

## Statistical analysis

Unless stated otherwise, significance was assessed by unpaired $t$ tests or ANOVA calculated with GraphPad Prism seven for Windows software. Significance values are shown as ****$p < 0.0001$, ***$p < 0.001$, **$p < 0.01$, *$p < 0.05$.

## Acknowledgments

We thank Thijn R Brummelkamp and Jacqueline Staring (Netherlands Cancer Institute) and Paul Helquist and Bruce Malencon (University of Notre Dame) for providing valuable reagents, and Michael Chua and Maryna Kapustina for expert technical advice. This work was supported in part by grants from the National Institute of Allergy and Infectious Disease (R01-AI103083 and R01-AI131685 to SML, and T32-AI007151 to EER-S). The funding agencies had no role in study design, data collection and interpretation, or the decision to submit the work for publication.

## Additional information

### Funding

| Funder | Grant reference number | Author |
|---|---|---|
| National Institute of Allergy and Infectious Diseases | R01-AI103083 | Stanley M Lemon |
| National Institute of Allergy and Infectious Diseases | R01-AI131685 | Stanley M Lemon |
| National Institute of Allergy and Infectious Diseases | T32-AI007151 | Efraín E Rivera-Serrano |

The funders had no role in study design, data collection and interpretation, or the decision to submit the work for publication.

### Author contributions

Efraín E Rivera-Serrano, Conceptualization, Data curation, Formal analysis, Investigation, Visualization, Methodology, Writing—original draft, Writing—review and editing, Designed and carried out all aspects of the experiments, collated and analyzed data, and wrote the manuscript; Olga González-López, Investigation, Methodology, Writing—review and editing, Carried out antibody neutralization assays and experiments with the HAV/FLuc sub-genomic replicon, and reviewed the manuscript; Anshuman Das, Investigation, Writing—review and editing, Developed the HAV-NanoLuc recombinant reporter virus, assisted with gradient purification of viruses and viral infection experiments, and reviewed the manuscript; Stanley M Lemon, Conceptualization, Supervision, Funding acquisition, Project administration, Writing—review and editing, Designed experiments, wrote the manuscript, and supervised the study

### Author ORCIDs

Efraín E Rivera-Serrano (iD) http://orcid.org/0000-0002-1039-7182
Stanley M Lemon (iD) http://orcid.org/0000-0003-1450-806X

### Decision letter and Author response

Decision letter https://doi.org/10.7554/eLife.43983.032
Author response https://doi.org/10.7554/eLife.43983.033

## Additional files

### Supplementary files

• Supplementary file 1. Nucleotide sequences of sgRNAs, gene-specific RT-PCR primers and siRNAs.
DOI: https://doi.org/10.7554/eLife.43983.029

• Transparent reporting form
DOI: https://doi.org/10.7554/eLife.43983.030

### Data availability

All data generated or analysed during this study are included in the manuscript and supplement files. Source data tables have been provided for Figures 1-5 and Figure 1-figure supplement 3, Figure 2-figure supplement 5, Figure 3-figure supplement 2, Figure 4-figure supplement 1B, Figure 5-figure supplement 1A, Figure 5-figure supplement 2A and Figure 5-figure supplement 2B.

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
