## [Decision Letter]

Thank you for submitting your article "Cellular entry and uncoating of naked and quasi-enveloped human hepatoviruses" for consideration by *eLife*. Your article has been reviewed by three peer reviewers, and the evaluation has been overseen by Wes Sundquist as a Reviewing Editor and Wenhui Li as the Senior Editor. The following individuals involved in review of your submission have agreed to reveal their identity: Joerg Votteler and Mary Estes.

The reviewers have discussed the reviews with one another and the Reviewing Editor has drafted this decision to help you prepare a revised submission.

General Assessment

This manuscript describes how two different forms of HAV, naked capsids (HAV) and quasi-enveloped virions (eHAV), enter cells. Using multiple complementary techniques (inhibitors, siRNA depletion experiments, immunofluorescence, membrane permeability assays), the authors demonstrate that both virion types enter cells primarily by clathrin- and dynamin-dependent endocytosis, facilitated by the β1 integrin. Virion entry differs in that naked HAV uncoats more rapidly in late endosomes, whereas, eHAV traffics to lysosomes in a pathway that requires the host factor ALIX. In the lysosome, the quasi-envelope of eHAV is enzymatically degraded for particle uncoating and subsequent release from the endolysosomal membrane, in a reaction that induces lysosomal permeability. The entry of both HAV forms is mechanistically different from other members of the Picornaviridae family.

Overall this is an important contribution that demonstrates that the entry pathways of naked HAV and eHAV differ in important ways, and expands our understanding of our quasi-enveloped viruses enter and infect cells.

There are no major issues that must be addressed prior to publication.

Issues for the authors' consideration:

Do the authors know if the kinetics of HAV and eHAV delivery to the late endosome and lysosome, respectively, parallel other cellular cargos? The times provided, 1-2 hrs and ~ 4hrs, seem quite long even accounting for "uncoating" processes, because fluid phase cargos typically reach lysosomes within 20 minutes or less. Any thoughts on why the viruses seem to be requiring longer times?

Have the authors co-infected Huh7.5 cells with both HAV and eHAV? For instance, they could detect the naked HAV using the K34C8 antibody and the eHAV with a fluorescent fixable membrane. Do they enter in the same clathrin coated pits, reach the same late endosomes? At what junction is the ALIX-based sorting taking place?

The discussion of the involvement of ALIX and the ESCRT pathway (data from Figure 2) is confusing. ESCRT proteins, including ALIX, are cytoplasmic (and are involved in sorting cargo proteins into vesicles that bud into the endosome). It is hard to imagine how these processes would affect eHAV trafficking directly, and this should be noted. It is, of course, possible that the ALIX requirement reflects changes in endosomal maturation (as is also suggested), but the discussion should be clarified to note that it is difficult to see how ALIX would come into direct contact with eHAV.

Many ESCRT-III proteins like CHMP2A have homologues (CHMP2B), and their function is often redundant. This might explain why CHMP2A knockdown has no effect.

Figures 3/4. Endocytosed eHAV capsids are still inside a membrane envelope at ~6 hpi (Figure 3D). Endocytosed eHAV capsids are stable at 4 hpi, but starting to uncoat 6 hpi (Figure 4A). Can quantitative analyses of the microscopy data determine, which step comes first, degradation of the membrane or capsid uncoating?

Figures 3E,4E: how many structures in how many cells were analyzed?

Figure 4D and Figure 4—figure supplement 1B: the authors state that the reduction in protein synthesis in cells infected with eHAV is comparable with cells infected with HRV14. I don't think the data really show that, especially in supplementary Figure 4B. The HRV14 effects seem to be much stronger (but the virus was quantified in a different way, which makes it difficult to compare).

The (negative) data presented in Figure 5 showing that PLA2G16 is dispensable for HAV entry could be condensed or moved to the supplements.

Figure 5E should be modified to show naked HAV going to late endosomes. How naked HAV escapes these endosomes also remains unclear and that could be discussed in more depth. This paper currently focuses more on the quasi-enveloped virus.

Abstract and elsewhere in the manuscript: the authors state that eHAV virions enter cells via a distinct endocytic pathway. However, this language could be clarified since the general entry pathway (β1-integrin-dependent clathrin-mediated endocytosis seems to be similar (and perhaps identical, albeit with differences in the mode of β1-integrin binding), and the major differences seem to be when (and perhaps how) naked HAV and eHAV exit the endolysosomal compartments.

It would be helpful if the authors could comment on the purity of the preparations of naked HAV versus eHAV? While these different preparations gave unique results (and the earlier results in their initial Nature paper show that the two species can be well separated on an iodixanol gradient), it would be helpful if information on how the preparations used in this study were characterized.

Some data suggest that poliovirus expands as part of the entry process. Is it possible this is also true for HAV and does it merit discussion?

---

## [Author Response]

Issues for the authors' consideration:Do the authors know if the kinetics of HAV and eHAV delivery to the late endosome and lysosome, respectively, parallel other cellular cargos? The times provided, 1-2 hrs and ~ 4hrs, seem quite long even accounting for "uncoating" processes, because fluid phase cargos typically reach lysosomes within 20 minutes or less. Any thoughts on why the viruses seem to be requiring longer times?

We thank the reviewers for highlighting this facet of hepatovirus entry. We have not done formal comparative studies with fluid phase cargos, but we agree that the intracellular trafficking of hepatoviruses to their destination is likely to be slower. However, it might not be much slower than for other viruses. We found both HAV and eHAV capsid antigen were associated with Rab7a^+^ compartments by 1-2 hrs, and that 4 hrs were required for eHAV to reach LAMP1^+^ and VAMP8^+^ lysosomes. By contrast, in carefully done kinetic studies, R.M. Mingo et al. (J Virol 89:2931, 2015) reported that 1-2 hrs were required for Ebola virus (EBOV) and SARS coronavirus virus-like particles (VLPs) to reach NPC1^+^ late endolysosomes. In addition, M. Huttenen et al. reported that coxsackievirus A9 (like HAV, a picornavirus) did not uncoat in neutral multivesicular endosomes until about 2 hrs postinfection (J Virol 88:5138, 2014). We have no good explanation for why the trafficking of hepatoviruses might be somewhat slower than these other viruses. It is not a phenomenon specific to the hepatocyte-derived cells we used, as many of the results in this manuscript were reproduced in HeLa and U2OS cells with similar kinetics (data not shown). It is possible that hepatoviruses engage atypical sorting signals or must undergo specific modifications during trafficking that impact their sorting kinetics.

Have the authors co-infected Huh7.5 cells with both HAV and eHAV? For instance, they could detect the naked HAV using the K34C8 antibody and the eHAV with a fluorescent fixable membrane. Do they enter in the same clathrin coated pits, reach the same late endosomes? At what junction is the ALIX-based sorting taking place?

Whether eHAV and HAV enter into the same clathrin-coated pits is an interesting question. We believe this is likely, but we have not formally demonstrated it. The experiments we report in the manuscript were done in parallel with the two virion types, and the cells were fixed and processed at almost identical times. Under these conditions, the initial steps in endosomal trafficking were indistinguishable for naked HAV and quasi-enveloped eHAV. Both the degree and kinetic of co-localization of capsid with clathrin, integrin β1, and the endosomal markers were similar for HAV and eHAV in multiple independent experiments, suggesting to us that they undergo endocytosis through similar clathrin-coated pits and travel through the same early/late endosomes – despite the fact that the specific ligands involved on the virion surfaces must be different. We did carry out the experiment suggested here involving co-infection of cells with gradient-purified naked HAV and PKH26-labelled eHAV virions. However, our ability to interpret the results of the experiment was confounded by our inability to absolutely distinguish between naked virions and eHAV particles that had lost their membrane due to the saponin treatment. (This was not an issue in the experiments shown in Figure 3D,E, in which we quantified only PKH26^+^ versus PKH26^-^ HAV puncta co-localizing with LAMP1, since naked HAV never trafficked to lysosomes.) With regard to the second part of this question, our data suggest that the ALIX-dependent sorting of eHAV to the lysosome diverges from trafficking of the naked virion at the late endosome stage. Naked HAV uncoats in the late endosome, but eHAV requires further trafficking. A simple, speculative explanation for this is that an essential receptor that binds to and acts as a trigger to initiate uncoating of the capsid is expressed in late, but not early endosomes, and that the naked particle uncoats when it first encounters it. The membrane occluding the capsid in the eHAV virion precludes this trigger from interacting with the capsid until eHAV has trafficked all the way to lysosomes, and the membrane is degraded by the actions of lysosomal acid lipase and NPC1.

The discussion of the involvement of ALIX and the ESCRT pathway (data from Figure 2) is confusing. ESCRT proteins, including ALIX, are cytoplasmic (and are involved in sorting cargo proteins into vesicles that bud into the endosome). It is hard to imagine how these processes would affect eHAV trafficking directly, and this should be noted. It is, of course, possible that the ALIX requirement reflects changes in endosomal maturation (as is also suggested), but the discussion should be clarified to note that it is difficult to see how ALIX would come into direct contact with eHAV.

We thank the referees for pointing this out, and agree completely with the need for clarification upon reviewing our original text. We have modified the Discussion to state that “ALIX mediates sorting of eHAV to the lysosome prior to degradation of its membrane, and thus promotes the entry process at a point during which the VP2 YPX_3_L motifs are occluded by the quasi-envelope, and moreover are within the lumen of the endosome and not available to interact with cytoplasmic ALIX. Rather than a direct interaction with the virus, the requirement for ALIX in eHAV entry is more likely to reflect a role for the ESCRT-associated protein in maturation and trafficking of the late endosome, akin to its role in entry of human papillomavirus and arenaviruses (Gräßel et al., 2016; Pasqual et al., 2011).”

Many ESCRT-III proteins like CHMP2A have homologues (CHMP2B), and their function is often redundant. This might explain why CHMP2A knockdown has no effect.

We agree that functional redundancy among ESCRT-III homologs could explain the apparent lack of a requirement for these proteins. However, the degree of CHMP2A mRNA depletion was substantially less than that we were able to achieve with siRNAs targeting ALIX (see Figure 2—figure supplement 5). We have modified the text in the Results section to mention both of these possibilities. It is interesting to note, however, that we have previously observed that depleting CHMP2A alone, as done here, resulted in a ~70% reduction in the release of quasi-enveloped eHAV from infected cells (McKnight et al., 2017).

Figures 3/4. Endocytosed eHAV capsids are still inside a membrane envelope at ~6 hpi (Figure 3D). Endocytosed eHAV capsids are stable at 4 hpi, but starting to uncoat 6 hpi (Figure 4A). Can quantitative analyses of the microscopy data determine, which step comes first, degradation of the membrane or capsid uncoating?

Degradation of the eHAV membrane and uncoating of the capsid appear to occur in close linkage temporally and we do not believe that it is possible to identify which comes first with any degree of confidence from imaging data alone. However, we did observe PKH26-negative puncta labelled with capsid antibody that colocalize with LAMP1 at ~8 hrs (Figure 3D), consistent with eHAV particles that have lost their membrane within the lysosome but retain an intact capsid. Rarely did we observe the opposite, PKH-positive puncta colocalizing with LAMP1 but not labeled with capsid antibody – and when these were observed it was not possible to distinguish them from nonspecific exosomes. Other, orthogonal data suggest strongly that membrane decay precedes uncoating of the eHAV capsid. We know from prior studies that the capsid (VP1pX specifically) is protected from proteinase K digestion when it is within the quasi-enveloped virion (Feng et al., 2013). Yet, anti-capsid antibodies quite potently neutralize eHAV infectivity within an endosomal compartment when added to cell cultures as late as 4 hrs post-infection – and even later when lysosomal acid lipase is inhibited (see Figure 3F). The simplest explanation reconciling these observations is that the eHAV membrane degrades prior to uncoating, allowing neutralizing antibody access to the capsid. If uncoating were to occur prior to membrane decay, we would not expect antibody to neutralize intracellular virus at a time consistent with lysosomal localization.

Figures 3E, 4E: how many structures in how many cells were analyzed?

For Figure 3E, we analyzed an average of 17.7 (DMSO) to 20.0 (Lalistat-2) structures per cell in each of 7 cells for each condition. These values are now included in the Figure 3-source data 1 file. The ratio (PKH26^+^/PKH26^-^) calculated for each individual cell is plotted in the figure. For Figure 4E, the number of dextran^+^ compartments in each cell is plotted on the y-axis, and ranged from a mean of 14.0 (mock-infected cells at 6 hrs) to 0.8 (eHAV-infected cells at 12 hrs). Seven to 10 cells were analyzed in each condition; the values for each cell are plotted individually in the figure and are included in the Figure 4—source data 1 file.

Figure 4D and Figure 4—figure supplement 1B: the authors state that the reduction in protein synthesis in cells infected with eHAV is comparable with cells infected with HRV14. I don't think the data really show that, especially in supplementary Figure 4B. The HRV14 effects seem to be much stronger (but the virus was quantified in a different way, which makes it difficult to compare).

We certainly agree, as this is what our quantification in Figure 4D demonstrated (~40% reduced with HAV vs. ~60% reduced with HRV14), and we also agree that comparing between the two virus types is complicated technically. We have modified the text, and now describe the eHAV effect as “similar but not as strong” as rhinovirus 14. Importantly, as we describe in the manuscript, we saw reductions in protein synthesis only with eHAV and never with naked HAV in the ribotoxin assays.

The (negative) data presented in Figure 5 showing that PLA2G16 is dispensable for HAV entry could be condensed or moved to the supplements.

With respect, we disagree with this particular suggestion. Staring et al. (Nature, 541:412, 2017) recently identified PLA2G16 as a pan-picornaviral host factor required for successful transfer of the genomic RNA to the cytoplasm for translation of viral proteins following uncoating of endocytosed virus. Our data showing that PLA2G16 is NOT required during entry of either eHAV or HAV, and acts instead to restrict replication of the viral RNA distinguishes hepatoviral entry from entry of other picornaviruses. While somewhat philosophical, we consider these data to be positive (not negative), as they are indicative of an important biological difference in these viruses. Pharmacological agents targeting the role of PLA2G16 in enteroviral infections are currently being developed (Patent WO 2015/018797A3 Antiviral Compounds), enhancing the importance of this distinction.

Figure 5E should be modified to show naked HAV going to late endosomes. How naked HAV escapes these endosomes also remains unclear and that could be discussed in more depth. This paper currently focuses more on the quasi-enveloped virus.

We have modified Figure 5E as suggested, and also have expanded our discussion of the uncoating of the naked virion in the penultimate paragraph of the Discussion.

Abstract and elsewhere in the manuscript: the authors state that eHAV virions enter cells via a distinct endocytic pathway. However, this language could be clarified since the general entry pathway (β1-integrin-dependent clathrin-mediated endocytosis seems to be similar (and perhaps identical, albeit with differences in the mode of β1-integrin binding), and the major differences seem to be when (and perhaps how) naked HAV and eHAV exit the endolysosomal compartments.

We agree, and have modified the Abstract and the introductory paragraph of the Discussion to reflect that the early steps in entry of both virion types are quite similar.

It would be helpful if the authors could comment on the purity of the preparations of naked HAV versus eHAV? While these different preparations gave unique results (and the earlier results in their initial Nature paper show that the two species can be well separated on an iodixanol gradient), it would be helpful if information on how the preparations used in this study were characterized.

As described in the Materials and methods section, purified populations of quasi-enveloped and naked, nonenveloped hepatoviruses were generated by preparative isopycnic density gradient centrifugation using a protocol nearly identical to that described previously (Feng et al., 2013). Similar and unequivocal results were obtained using multiple batches of virus prepared in this fashion. The two populations of virions are separated at a very high level of resolution, with essentially no quasi-enveloped virus present in the high density, naked virus fractions. Loss of the membrane from purified eHAV virions, generating naked virus, is always a concern, but eHAV is quite stable under storage conditions. The 7-8 log_10_ difference we observed in immediate post-entry neutralization of HAV versus eHAV (Figure 3E, 1 h timepoints) provides strong assurance of the purity of the stocks generated.

Some data suggest that poliovirus expands as part of the entry process. Is it possible this is also true for HAV and does it merit discussion?

We now mention this in the Discussion (modified penultimate paragraph). There is simply not enough known about the process of HAV uncoating to predict whether its capsid also undergoes the irreversible expansion seen with poliovirus following interactions with its receptor. We have further modified the Discussion to point out some of the salient differences in the structures of these two picornaviruses that might be relevant to uncoating.